# Unsupervised Video Domain Adaptation for Action Recognition: A Disentanglement Perspective

**Pengfei Wei**
AI Lab, ByteDance
pengfei.wei@bytedance.com

**Lingdong Kong**
National University of Singapore
lingdong@comp.nus.edu.sg

**Xinghua Qu**
AI Lab, ByteDance
quxinghua17@gmail.com

**Yi Ren**
AI Lab, ByteDance
ren.yi@bytedance.com

**Zhiqiang Xu**
MBZUAI
zhiqiang.xu@mbzuai.ac.ae

**Jing Jiang**
University of Technology Sydney
Jing.Jiang@uts.edu.au

**Xiang Yin**
AI Lab, ByteDance
yinxiang.stephen@bytedance.com

## Abstract

Unsupervised video domain adaptation is a practical yet challenging task. In this work, for the first time, we tackle it from a disentanglement view. Our key idea is to handle the spatial and temporal domain divergence separately through disentanglement. Specifically, we consider the generation of cross-domain videos from two sets of latent factors, one encoding the static information and another encoding the dynamic information. A ***Transfer Sequential VAE (TranSVAE)*** framework is then developed to model such generation. To better serve for adaptation, we propose several objectives to constrain the latent factors. With these constraints, the spatial divergence can be readily removed by disentangling the static domain-specific information out, and the temporal divergence is further reduced from both frame- and video-levels through adversarial learning. Extensive experiments on the UCF-HMDB, Jester, and Epic-Kitchens datasets verify the effectiveness and superiority of *TranSVAE* compared with several state-of-the-art approaches.[1]

## 1  Introduction

Over the past decades, unsupervised domain adaptation (UDA) has attracted extensive research attention [41]. Numerous UDA methods have been proposed and successfully applied to various real-world applications, *e.g.*, object recognition [38, 42, 47], semantic segmentation [51, 19, 33, 24], and object detection [3, 15, 45]. However, most of these methods and their applications are limited to the image domain, while much less attention has been devoted to video-based UDA, where the latter is undoubtedly more challenging.

Compared with image-based UDA, the source and target domains also differ temporally in video-based UDA. Images are spatially well-structured data, while videos are sequences of images with both spatial and temporal relations. Existing image-based UDA methods can hardly achieve satisfactory performance on video-based UDA tasks as they fail to consider the temporal dependency of video frames in handling the domain gaps. For instance, in video-based cross-domain action recognition tasks, domain gaps are presented by not only the actions of different persons in different scenarios but also the actions that appear at different timestamps or last at different time lengths.

---

[1]Code is publicly available at: https://github.com/ldkong1205/TranSVAE

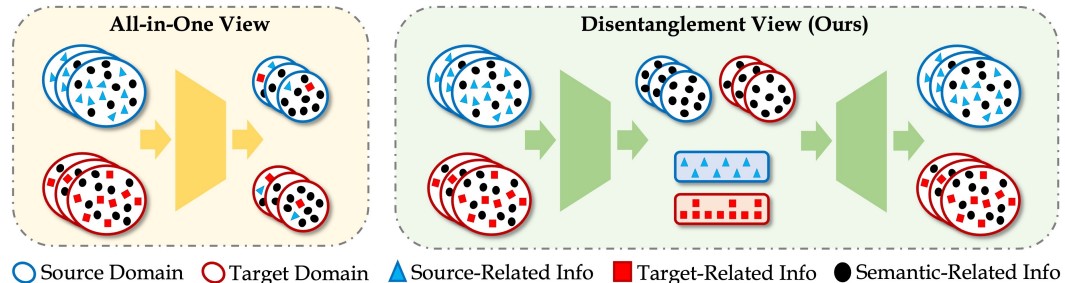

Source Domain ◯  Target Domain ◯  ▲ Source-Related Info  ■ Target-Related Info  ● Semantic-Related Info

Figure 1: Conceptual comparisons between the traditional *all-in-one view* and the proposed *disentanglement view*. Prior works often seek to compress implicit domain information to obtain domain-indistinguishable representations; while in this work, we pursue explicit decouplings of domain-specific information from other information via generative modeling.

Recently, few works have been proposed for video-based UDA. The key idea is to achieve *domain alignment* by aligning both frame- and video-level features through adversarial learning [7, 26], contrastive learning [37, 31], attention [10], or combination of these mechanisms, *e.g.*, adversarial learning with attention [29, 8]. Though they have advanced video-based UDA, there is still room for improvement. Generally, existing methods follow an *all-in-one* way, where both spatial and temporal domain divergence are handled together, for adaptation (Fig. 1, ▢). However, cross-domain videos are highly complex data containing diverse mixed-up information, *e.g.*, domain, semantic, and temporal information, which makes the simultaneous elimination of spatial and temporal divergence insufficient. This motivates us to handle the video-based UDA from a *disentanglement* perspective (Fig. 1, ▢) so that the spatial and temporal divergence can be well handled separately.

To achieve this goal, we first consider the *generation* process of cross-domain videos, as shown in Fig. 2, where a video is generated from two sets of latent factors: one set consists of a sequence of random variables, which are *dynamic* and incline to encode the semantic information for downstream tasks, *e.g.*, action recognition; another set is *static* and introduces some domain-related spatial information to the generated video, *e.g.*, style or appearance. Specifically, the blue / red nodes are the observed source / target videos $\mathbf{x}^{\mathcal{S}}$ / $\mathbf{x}^{\mathcal{T}}$, respectively, over $t$ timestamps. Static latent variables $\mathbf{z}_d^{\mathcal{S}}$ and $\mathbf{z}_d^{\mathcal{T}}$ follow a joint distribution and combining either of them with dynamic latent variables $\mathbf{z}_t$ constructs one video data of a domain.

With the above generative model, we develop a ***Trans**fer **S**equential **V**ariational **A**uto**E**ncoder* (*TranSVAE*) for video-based UDA. *TranSVAE* handles the cross-domain divergence in two levels, where the first level removes the spatial divergence by disentangling $\mathbf{z}_d$ from $\mathbf{z}_t$; while the second level eliminates the temporal divergence of $\mathbf{z}_t$. To achieve this, we leverage appropriate constraints to ensure that the disentanglement indeed serves the adaptation purpose. Firstly, we enable a good decoupling of the two sets of latent factors by minimizing their *mutual dependence*. This encourages these two latent factor sets to be mutually independent. We then consider constraining each latent factor set. For $\mathbf{z}_d^{\mathcal{D}}$ with $\mathcal{D} \in \{\mathcal{S}, \mathcal{T}\}$, we propose a *contrastive triplet loss* to make them static and domain-specific. This makes us readily handle spatial divergence by disentangling $\mathbf{z}_d^{\mathcal{D}}$ out. For $\mathbf{z}_t$, we propose to align them across domains at both frame and video levels through *adversarial learning* so as to further eliminate the temporal divergence. Meanwhile, as downstream tasks use $\mathbf{z}_t$ as input, we also add the *task-specific supervision* on $\mathbf{z}_t$ extracted from source data (*w/* ground-truth).

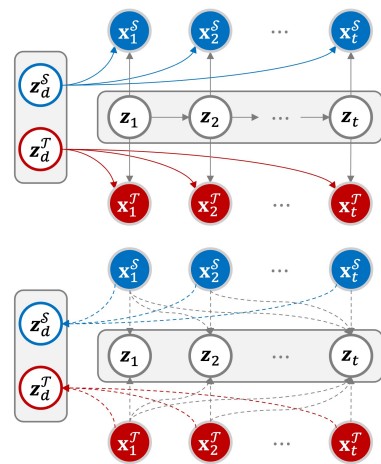

Figure 2: Graphical illustrations of the proposed *generative* (1st figure) and *inference* (2nd figure) models for video domain disentanglement.

To the best of our knowledge, this is the first work that tackles the challenging video-based UDA from a domain disentanglement view. We conduct extensive experiments on popular benchmarks (UCF-HMDB, Jester, Epic-Kitchens) and the results show that *TranSVAE* consistently outperforms previous state-of-the-art methods by large margins. We also conduct comprehensive ablation studies

and disentanglement analyses to verify the effectiveness of the latent factor decoupling. The main contribution of the paper is summarized as follows:

- We provide a generative perspective on solving video-based UDA problems. We develop a generative graphical model for the cross-domain video generation process and propose to utilize the sequential VAE as the base generative model.

- Based on the above generative view, we propose a *TranSVAE* framework for video-based UDA. By developing four constraints on the latent factors to enable disentanglement to benefit adaptation, the proposed framework is capable of handling the cross-domain divergence from both spatial and temporal levels.

- We conduct extensive experiments on several benchmark datasets to verify the effectiveness of *TranSVAE*. A comprehensive ablation study also demonstrates the positive effect of each loss term on video domain adaptation.

## 2 Related Work

**Unsupervised Video Domain Adaptation**. Despite the great progress in image-based UDA, only a few methods have recently attempted video-based UDA. In [7], a temporal attentive adversarial adaptation network (TA$^3$N) is proposed to integrate a temporal relation module for temporal alignment. Choi *et al.* [10] proposed a SAVA method using self-supervised clip order prediction and clip attention-based alignment. Based on a cross-domain co-attention mechanism, the temporal co-attention network TCoN [29] focused on common key frames across domains for better alignment. Luo *et al.* [26] pay more attention to the domain-agnostic classifier by using a network topology of the bipartite graph to model the cross-domain correlations. Instead of using adversarial learning, Sahoo *et al.* [31] developed an end-to-end temporal contrastive learning framework named CoMix with background mixing and target pseudo-labels. Recently, Chen *et al.* [8] learned multiple domain discriminators for multi-level temporal attentive features to achieve better alignment, while Turrisi *et al.* [37] exploited two-headed deep architecture to learn a more robust target classifier by the combination of cross-entropy and contrastive losses. Although these approaches have advanced video-based UDA tasks, they all adopted to align features with diverse information mixed up from a compression perspective, which leaves room for further improvements.

**Multi-Modal Video Adaptation**. Most recently, there are also a few works integrating multiple modality data for video-based UDA. Although we only use the single modality RGB features, we still discuss this multi-modal research line for a complete literature review. The very first work exploring the multi-modal nature of videos for UDA is MM-SADA [28], where the correspondence of multiple modalities was exploited as a self-supervised alignment in addition to adversarial alignment. A later work, spatial-temporal contrastive domain adaptation (STCDA) [34], utilized a video-based contrastive alignment as the multi-modal domain metric to measure the video-level discrepancy across domains. [18] proposed cross-modal and cross-domain contrastive losses to handle feature spaces across modalities and domains. [43] leveraged both cross-modal complementary and cross-modal consensus to learn the most transferable features through a CIA framework. In [12], the authors proposed to generate noisy pseudo-labels for the target domain data using the source pre-trained model and select the clean samples in order to increase the quality of the pseudo-labels. Lin *et al.* [23] developed a cycle-based approach that alternates between spatial and spatiotemporal learning with knowledge transfer. Generally, all the above methods utilize the flow as the auxiliary modality input. Recently, there are also methods exploring other modalities, for instance, A3R [48] with audios and MixDANN [44] with wild data. It is worth noting that the proposed *TranSVAE* – although only uses single modality RGB features – surprisingly achieves better UDA performance compared with most current state-of-the-art multi-modal methods, which highlights our superiority.

**Disentanglement**. Feature disentanglement is a wide and hot research topic. We only focus on works that are closely related to ours. In the image domain, some works consider adaptation from a generative view. [4] learned a disentangled semantic representation across domains. A similar idea is then applied to graph domain adaptation [5] and domain generalization [16]. [13] proposed a novel informative feature disentanglement, equipped with the adversarial network or the metric discrepancy model. Another disentanglement-related topic is sequential data generation. To generate videos, existing works [22, 50, 1] extended VAE to a recurrent form with different recursive structures. In

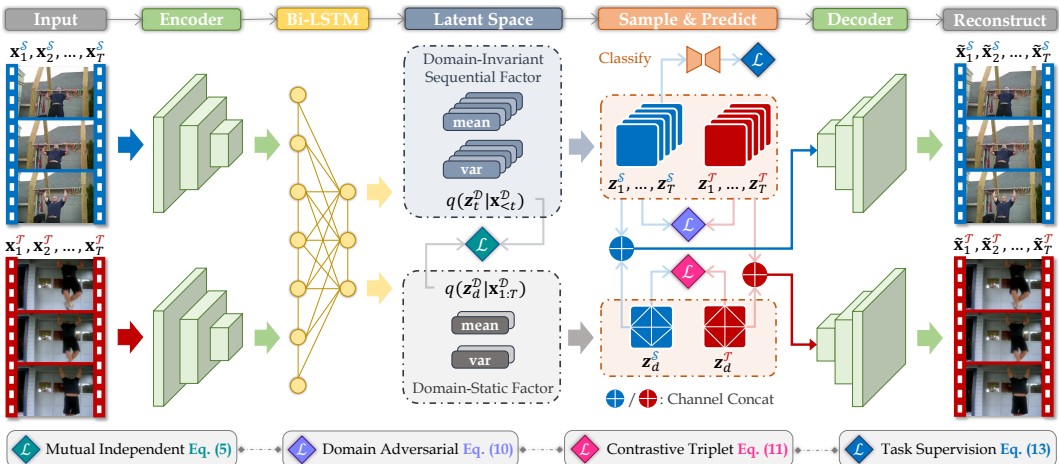

Figure 3: *TranSVAE* overview. The input videos are fed into an encoder to extract the visual features, followed by an LSTM to explore the temporal information. Two groups of *mean* and *variance* networks are then applied to model the posterior of the latent factors, *i.e.*, $q(\mathbf{z}_t^{\mathcal{D}}|\mathbf{x}_{<t}^{\mathcal{D}})$ and $q(\mathbf{z}_d^{\mathcal{D}}|\mathbf{x}_{1:T}^{\mathcal{D}})$. The new representations $\mathbf{z}_1^{\mathcal{D}}, ..., \mathbf{z}_T^{\mathcal{D}}$ and $\mathbf{z}_d^{\mathcal{D}}$ are sampled, and then concatenated and passed to a decoder for reconstruction. Four constraints are proposed to regulate the latent factors for adaptation.

this paper, we present a VAE-based structure to generate cross-domain videos. We aim at tackling video-based UDA from a new perspective: sequential domain disentanglement and transfer.

## 3   Technical Approach

Formally, for a typical video-based UDA problem, we have a source domain $\mathcal{S}$ and a target domain $\mathcal{T}$. Domain $\mathcal{S}$ contains sufficient labeled data, denoted as $\{(\mathbf{V}_i^{\mathcal{S}}, y_i^{\mathcal{S}})\}_{i=1}^{N_{\mathcal{S}}}$, where $\mathbf{V}_i^{\mathcal{S}}$ is a video sequence and $y_i^{\mathcal{S}}$ is the class label. Domain $\mathcal{T}$ consists of unlabeled data, denoted as $\{\mathbf{V}_i^{\mathcal{T}}\}_{i=1}^{N_{\mathcal{T}}}$. For a sequence $\mathbf{V}_i^{\mathcal{D}}$ from domain $\mathcal{D} \in \{\mathcal{S}, \mathcal{T}\}$, it contains $T$ frames $\{\mathbf{x}_{i\_1}^{\mathcal{D}}, ..., \mathbf{x}_{i\_T}^{\mathcal{D}}\}$ in total[2]. We further denote $N = N^{\mathcal{S}} + N^{\mathcal{T}}$. Domains $\mathcal{S}$ and $\mathcal{T}$ are of different distributions but share the same label space. The objective is to utilize both $\{(\mathbf{V}_i^{\mathcal{S}}, y_i^{\mathcal{S}})\}_{i=1}^{N_{\mathcal{S}}}$ and $\{\mathbf{V}_i^{\mathcal{T}}\}_{i=1}^{N_{\mathcal{T}}}$ to train a good classifier for domain $\mathcal{T}$. We present a table to list the notations in the Appendix.

**Framework Overview**. We adopt a VAE-based structure to model the cross-domain video generation process and propose to regulate the two sets of latent factors for adaptation purposes. The reconstruction process is based on the two sets of latent factors, *i.e.*, $\mathbf{z}_1^{\mathcal{D}}, ..., \mathbf{z}_T^{\mathcal{D}}$ and $\mathbf{z}_d^{\mathcal{D}}$ that are sampled from the posteriors $q(\mathbf{z}_t^{\mathcal{D}}|\mathbf{x}_{<t}^{\mathcal{D}})$ and $q(\mathbf{z}_d^{\mathcal{D}}|\mathbf{x}_{1:T}^{\mathcal{D}})$, respectively. The overall architecture consists of five segments including the encoder, LSTM, latent spaces, sampling, and decoder, as shown in Fig. 3. The vanilla structure only gives an arbitrary disentanglement of latent factors. To make the disentanglement facilitate adaptation, we carefully constrain the latent factors as follows.

**Video Sequence Reconstruction**. The overall architecture of *TranSVAE* follows a VAE-based structure [22, 50, 1] with two sets of latent factors $\mathbf{z}_1^{\mathcal{D}}, ..., \mathbf{z}_T^{\mathcal{D}}$ and $\mathbf{z}_d^{\mathcal{D}}$. The generative and inference graphical models are presented in Fig. 2. Similar to the conventional VAE, we use a standard Gaussian distribution for static latent factors. For dynamic ones, we use a sequential prior $\mathbf{z}_t^{\mathcal{D}}|\mathbf{z}_{<t}^{\mathcal{D}} \sim \mathcal{N}(\boldsymbol{\mu}_t, diag(\boldsymbol{\sigma}_t^2))$, that is, the prior distribution of the current dynamic factor is conditioned on the historical dynamic factors. The distribution parameters can be re-parameterized as a recurrent network, *e.g.*, LSTM, with all previous dynamic latent factors as the input. Denoting $\mathbf{Z}^{\mathcal{D}} = \{\mathbf{z}_1^{\mathcal{D}}, ..., \mathbf{z}_T^{\mathcal{D}}\}$ for simplification, we then get the prior as follows:

$$p(\mathbf{z}_d^{\mathcal{D}}, \mathbf{Z}^{\mathcal{D}}) = p(\mathbf{z}_d^{\mathcal{D}}) \prod_{t=1}^{T} p(\mathbf{z}_t^{\mathcal{D}}|\mathbf{z}_{<t}^{\mathcal{D}}). \tag{1}$$

---

[2]Without confusion, we omit the subscript $i$ for $\mathbf{V}^{\mathcal{D}}$ and the corresponding notations, *e.g.*, $\{\mathbf{x}_1^{\mathcal{D}}, ..., \mathbf{x}_T^{\mathcal{D}}\}$.

Following Fig. 2 (the 1st subfigure), $\mathbf{x}_t^{\mathcal{D}}$ is generated from $\mathbf{z}_d^{\mathcal{D}}$ and $\mathbf{z}_t^{\mathcal{D}}$, and we thus model $p(\mathbf{x}_t^{\mathcal{D}}|\mathbf{z}_d^{\mathcal{D}}, \mathbf{z}_t^{\mathcal{D}}) = \mathcal{N}(\boldsymbol{\mu}_t', diag(\boldsymbol{\sigma}_t'^2))$. The distribution parameters are re-parameterized by the decoder which can be flexible networks like the deconvolutional neural network. Using $\mathbf{V}^{\mathcal{D}} = \{\mathbf{x}_1^{\mathcal{D}}, ..., \mathbf{x}_T^{\mathcal{D}}\}$, the *generation* can be formulated as follows:

$$p(\mathbf{V}^{\mathcal{D}}) = p(\mathbf{z}_d^{\mathcal{D}}) \prod_{t=1}^{T} p(\mathbf{x}_t^{\mathcal{D}}|\mathbf{z}_d^{\mathcal{D}}, \mathbf{z}_t^{\mathcal{D}})p(\mathbf{z}_t^{\mathcal{D}}|\mathbf{z}_{<t}^{\mathcal{D}}). \tag{2}$$

Following Fig. 2 (the 2nd subfigure), we model the posterior distributions of the latent factors as another two Gaussian distributions, *i.e.*, $q(\mathbf{z}_d^{\mathcal{D}}|\mathbf{V}^{\mathcal{D}}) = \mathcal{N}(\boldsymbol{\mu}_d, diag(\boldsymbol{\sigma}_d^2))$ and $q(\mathbf{z}_t^{\mathcal{D}}|\mathbf{x}_{<t}^{\mathcal{D}}) = \mathcal{N}(\boldsymbol{\mu}_t'', diag(\boldsymbol{\sigma}_t''^2))$. The parameters of these two distributions are re-parameterized by the encoder, which can be a convolutional or LSTM module. However, the network of the static latent factors uses the whole sequence as the input while that of the dynamic latent factors only uses previous frames. Then the *inference* can be factorized as:

$$q(\mathbf{z}_d^{\mathcal{D}}, \mathbf{Z}^{\mathcal{D}}|\mathbf{V}^{\mathcal{D}}) = q(\mathbf{z}_d^{\mathcal{D}}|\mathbf{V}^{\mathcal{D}}) \prod_{t=1}^{T} q(\mathbf{z}_t^{\mathcal{D}}|\mathbf{x}_{<t}^{\mathcal{D}}). \tag{3}$$

Combining the above generation and inference, we obtain the VAE-related objective function as:

$$\mathcal{L}_{\text{svae}} = \mathbb{E}_{q(\mathbf{z}_d^{\mathcal{D}}, \mathbf{Z}^{\mathcal{D}}|\mathbf{V}^{\mathcal{D}})}[-\sum_{t=1}^{T} \log p(\mathbf{x}_t^{\mathcal{D}}|\mathbf{z}_d^{\mathcal{D}}, \mathbf{z}_t^{\mathcal{D}})]+$$

$$\text{KL}(q(\mathbf{z}_d^{\mathcal{D}}|\mathbf{V}^{\mathcal{D}})||p(\mathbf{z}_d^{\mathcal{D}})) + \sum_{t=1}^{T} \text{KL}(q(\mathbf{z}_t^{\mathcal{D}}|\mathbf{x}_{<t}^{\mathcal{D}})||p(\mathbf{z}_t^{\mathcal{D}}|\mathbf{z}_{<t}^{\mathcal{D}})), \tag{4}$$

which is a frame-wise negative variational lower bound. Only using the above vanilla VAE-based loss cannot guarantee that the disentanglement serves for adaptation, and thus we propose additional constraints on the two sets of latent factors.

**Mutual Dependence Minimization** (Fig. 3, ■). We first consider explicitly enforcing the two sets of latent factors to be mutually independent. To do so, we introduce the mutual information [2] loss $\mathcal{L}_{\text{mi}}$ to regulate the two sets of latent factors. Thus, we obtain:

$$\mathcal{L}_{\text{mi}}(\mathbf{z}_d^{\mathcal{D}}, \mathbf{Z}^{\mathcal{D}}) = \sum_{t=1}^{T} \text{KL}\left(q(\mathbf{z}_d^{\mathcal{D}}, \mathbf{z}_t^{\mathcal{D}})||q(\mathbf{z}_d^{\mathcal{D}})q(\mathbf{z}_t^{\mathcal{D}})\right) = \sum_{t=1}^{T}[H(\mathbf{z}_d^{\mathcal{D}}) + H(\mathbf{z}_t^{\mathcal{D}}) - H(\mathbf{z}_d^{\mathcal{D}}, \mathbf{z}_t^{\mathcal{D}})]. \tag{5}$$

To calculate Eq. (5), we need to estimate the densities of $\mathbf{z}_d^{\mathcal{D}}, \mathbf{z}_t^{\mathcal{D}}$ and $(\mathbf{z}_d^{\mathcal{D}}, \mathbf{z}_t^{\mathcal{D}})$. Following the non-parametric way in [9], we use the mini-batch weighted sampling as follows:

$$H(\mathbf{z}) = -\mathbb{E}_{q(\mathbf{z})}[\log q(\mathbf{z})] \approx -\log \frac{1}{M} \sum_{i=1}^{M}[\log \frac{1}{MN} \sum_{j=1}^{M} q\left(\mathbf{z}(\mathbf{x}_i)|\mathbf{x}_j\right)], \tag{6}$$

where $\mathbf{z}$ is $\mathbf{z}_d^{\mathcal{D}}, \mathbf{z}_t^{\mathcal{D}}$ or $(\mathbf{z}_d^{\mathcal{D}}, \mathbf{z}_t^{\mathcal{D}})$, $N$ denotes the data size and $M$ is the mini-batch size.

**Domain Specificity & Static Consistency** (Fig. 3, ■). A characteristic of the domain specificity, e.g., the video style or the objective appearance, is its static consistency over dynamic frames. With this observation, we enable the static and domain-specific latent factors so that we can remove the spatial divergence by disentangling them out. Mathematically, we hope that $\mathbf{z}_d^{\mathcal{D}}$ does not change a lot when $\mathbf{z}_t^{\mathcal{D}}$ varies over time. To achieve this, given a sequence, we randomly shuffle the temporal order of frames to form a new sequence. The static latent factors disentangled from the original sequence and the shuffled sequence should be ideally equal or be very close at least. This motivates us to minimize the distance between these two static factors. Meanwhile, to further enhance the domain specificity, we enforce the dynamic latent factors from different domains to have a large distance. To this end, we propose the following contrastive triplet loss:

$$\mathcal{L}_{\text{ctc}} = \max\left(D(\mathbf{z}_d^{\mathcal{D}^+}, \widetilde{\mathbf{z}}_d^{\mathcal{D}^+}) - D(\mathbf{z}_d^{\mathcal{D}^+}, \mathbf{z}_d^{\mathcal{D}^-}) + \mathbf{m}, 0\right), \tag{7}$$

where $D(\cdot, \cdot)$ is Euclidean distance, $\mathbf{m}$ is a margin set to 1 in the experiments, $\mathbf{z}_d^{\mathcal{D}^+}, \widetilde{\mathbf{z}}_d^{\mathcal{D}^+}$, and $\mathbf{z}_d^{\mathcal{D}^-}$ are static latent factors of the anchor sequence from domain $\mathcal{D}^+$, the shuffled sequence, and a randomly selected sequence from domain $\mathcal{D}^-$, respectively. $\mathcal{D}^+$ and $\mathcal{D}^-$ represent two different domains.

**Domain Temporal Alignment** (Fig. 3, ▢). We now consider to reduce the temporal divergence of the *dynamic* latent factors. There are several ways to achieve this, and in this paper, we take advantage of the most popular adversarial-based idea [14]. Specifically, we build a domain classifier to discriminate whether the data is from $\mathcal{S}$ or $\mathcal{T}$. When back-propagating the gradients, a gradient reversal layer (GRL) is adopted to invert the gradients. Like existing video-based UDA methods, we also conduct both frame-level and video-level alignments. Moreover, as TA³N [7] does, we exploit the temporal relation network (TRN) [49] to discover the temporal relations among different frames, and then aggregate all the temporal relation features into the final video-level features. This enables another level of alignment on the temporal relation features. Thus, we have:

$$\mathcal{L}_f = \frac{1}{N} \sum_{i=1}^{N} \frac{1}{T} \sum_{t=1}^{T} CE \left[ G_f(\mathbf{z}_{i\_t}^{\mathcal{D}}), d_i \right], \tag{8}$$

$$\mathcal{L}_r = \frac{1}{N} \sum_{i=1}^{N} \frac{1}{T-1} \sum_{n=2}^{T} CE \left[ G_r \left( TrN_n(\mathbf{Z}_i^{\mathcal{D}}) \right), d_i \right], \tag{9}$$

$$\mathcal{L}_v = \frac{1}{N} \sum_{i=1}^{N} CE \left[ G_v \left( \frac{1}{T-1} \sum_{n=2}^{T} TrN_n(\mathbf{Z}_i^{\mathcal{D}}) \right), d_i \right], \tag{10}$$

where $d_i$ is the domain label, $CE$ denotes the cross-entropy loss function, $\mathbf{Z}_i^{\mathcal{D}} = \{\mathbf{z}_{i\_1}^{\mathcal{D}}, ..., \mathbf{z}_{i\_T}^{\mathcal{D}}\}$, $TrN_i$ is the $n$-frame temporal relation network, $G_f$, $G_r$, and $G_v$ are the frame feature level, the temporal relation feature level, and the video feature level domain classifiers, respectively. To this end, we obtain the domain adversarial loss by summing up Eqs. (8-10):

$$\mathcal{L}_{\text{adv}} = \mathcal{L}_f + \mathcal{L}_r + \mathcal{L}_v. \tag{11}$$

We assign equal importance to these three levels of losses to reduce the overhead of the hyperparameter search. To this end, with $\mathcal{L}_{\text{mi}}$, $\mathcal{L}_{\text{ctc}}$, and $\mathcal{L}_{\text{adv}}$, the learned dynamic latent factors are expected to be domain-invariant (the three constraints are interactive and complementary to each other for obtaining the domain-invariant dynamic latent factor), and then can be used for downstream UDA tasks. In this paper, we specifically focus on action recognition as the downstream task.

**Task Specific Supervision** (Fig. 3, ▮). We further encourage the dynamic latent factors to carry the semantic information. Considering that the source domain has sufficient labels, we accordingly design the task supervision as the regularization imposed on $\mathbf{z}_i^{\mathcal{S}}$. This gives us:

$$\mathcal{L}_{\text{cls}} = \frac{1}{N^{\mathcal{S}}} \sum_{i=1}^{N^{\mathcal{S}}} \mathcal{L} \left( \mathcal{F}(\mathbf{Z}_i^{\mathcal{S}}), y_i^{\mathcal{S}} \right), \tag{12}$$

where $\mathcal{F}(\cdot)$ is a feature transformer mapping the frame-level features to video-level features, specifically a TRN in this paper, and $\mathcal{L}(\cdot, \cdot)$ is either cross-entropy or mean squared error loss according to the targeted task.

Although the dynamic latent factors are constrained to be domain-invariant, we do not completely rely on source semantics to learn features discriminative for the target domain. We propose to incorporate target pseudo-labels in task-specific supervision. During the training, we use the prediction network obtained in the previous epoch to generate the target pseudo-labels of the unlabelled target training data for the current epoch. However, to increase the reliability of target pseudo-labels, we let the prediction network be trained only on the source supervision for several epochs and then integrate the target pseudo-labels in the following training epochs. Meanwhile, a confidence threshold is set to determine whether to use the target pseudo-labels or not. Thus, we have the final task-specific supervision as follows:

$$\mathcal{L}_{\text{cls}} = \frac{1}{N} \left( \sum_{i=1}^{N^{\mathcal{S}}} \mathcal{L}(\mathcal{F}(\mathbf{Z}_i^{\mathcal{S}}), y_i^{\mathcal{S}}) + \sum_{i=1}^{N^{\mathcal{T}}} \mathcal{L}(\mathcal{F}(\mathbf{Z}_i^{\mathcal{T}}), \widetilde{y}_i^{\mathcal{T}}) \right), \tag{13}$$

where $\widetilde{y}_i^{\mathcal{T}}$ is the pseudo-label of $\mathbf{Z}_i^{\mathcal{T}}$.

**Summary**. To this end, we reach the final objective function of our *TranSVAE* framework as follows:

$$\mathcal{L}' = \mathcal{L}_{\text{svae}} + \lambda_1 \mathcal{L}_{\text{mi}} + \lambda_2 \mathcal{L}_{\text{adv}} + \lambda_3 \mathcal{L}_{\text{ctc}} + \lambda_4 \mathcal{L}_{\text{cls}}, \tag{14}$$

where $\lambda_i$ with $i = 1, 2, 3, 4$ denotes the loss balancing weight.

# 4 Experiments

In this section, we conduct extensive experimental studies on popular video-based UDA benchmarks to verify the effectiveness of the proposed *TranSVAE* framework.

## 4.1 Datasets

**UCF-HMDB** is constructed by collecting the relevant and overlapping action classes from $UCF_{101}$ [35] and $HMDB_{51}$ [20]. It contains 3,209 videos in total with 1,438 training videos and 571 validation videos from $UCF_{101}$, and 840 training videos and 360 validation videos from $HMDB_{51}$. This in turn establishes two video-based UDA tasks: $\mathbf{U} \to \mathbf{H}$ and $\mathbf{H} \to \mathbf{U}$.

**Jester** [27] consists of 148,092 videos of humans performing hand gestures. Pan *et al.* [29] constructed a large-scale cross-domain benchmark with seven gesture classes, and form a single transfer task $\mathbf{J}_{\mathcal{S}} \to \mathbf{J}_{\mathcal{T}}$, where $\mathbf{J}_{\mathcal{S}}$ and $\mathbf{J}_{\mathcal{T}}$ contain 51,498 and 51,415 video clips, respectively.

**Epic-Kitchens** [11] is a challenging egocentric dataset consisting of videos capturing daily activities in kitchens. [28] constructs three domains across the eight largest actions. They are $\mathbf{D}_1$, $\mathbf{D}_2$, and $\mathbf{D}_3$ corresponding to P08, P01, and P22 kitchens of the full dataset, resulting in six cross-domain tasks.

**Sprites** [22] contains sequences of animated cartoon characters with 15 action categories. The appearances of characters are fully controlled by four attributes, *i.e.*, body, top wear, bottom wear, and hair. We construct two domains, $\mathbf{P}_1$ and $\mathbf{P}_2$. $\mathbf{P}_1$ uses the *human* body with attributes randomly selected from 3 top wears, 4 bottom wears, and 5 hairs, while $\mathbf{P}_2$ uses the *alien* body with attributes randomly selected from 4 top wears, 3 bottom wears, and 5 hairs. The attribute pools are non-overlapping across domains, resulting in completely heterogeneous $\mathbf{P}_1$ and $\mathbf{P}_2$. Each domain has 900 video sequences, and each sequence is with 8 frames.

## 4.2 Implementation Details

**Architecture**. Following the latest works [31, 37], we use I3D [6] as the backbone[3]. However, different from CoMix which jointly trains the backbone, we simply use the pretrained I3D model on Kinetics [17], provided by [6], to extract RGB features. For the first three benchmarks, RGB features are used as the input of *TranSVAE*. For Sprites, we use the original image as the input, for the purpose of visualizing the reconstruction and disentanglement results. We use the shared encoder and decoder structures across the source and target domains. For RGB feature inputs, the encoder and decoder are fully connected layers. For original image inputs, the encoder and decoder are the convolution and deconvolution layers (from DCGAN [46]), respectively. For the TRN model, we directly use the one provided by [7]. Other details on this aspect are placed in Appendix.

**Configurations**. Our *TranSVAE* is implemented with PyTorch [30]. We use Adam with a weight decay of $1e^{-4}$ as the optimizer. The learning rate is initially set to be $1e^{-3}$ and follows a commonly used decreasing strategy in [14]. The batch size and the learning epoch are uniformly set to be 128 and 1,000, respectively, for all the experiments. We uniformly set 100 epochs of training under only source supervision and involved the target pseudo-labels afterward. Following the common protocol in video-based UDA [37], we perform hyperparameter selection on the validation set. The specific hyperparameters used for each task can be found in the Appendix. NVIDIA A100 GPUs are used for all experiments. Kindly refer to our Appendix for all other details.

Table 1: UDA performance comparisons on UCF-HMDB.

| Method & Year | Backbone | $\mathbf{U} \to \mathbf{H}$ | $\mathbf{H} \to \mathbf{U}$ | Average ↑ |
|---|---|---|---|---|
| DANN (JMLR'16) | ResNet-101 | 75.28 | 76.36 | 75.82 |
| JAN (ICML'17) | ResNet-101 | 74.72 | 76.69 | 75.71 |
| AdaBN (PR'18) | ResNet-101 | 72.22 | 77.41 | 74.82 |
| MCD (CVPR'18) | ResNet-101 | 73.89 | 79.34 | 76.62 |
| TA$^3$N (ICCV'19) | ResNet-101 | 78.33 | 81.79 | 80.06 |
| ABG (MM'20) | ResNet-101 | 79.17 | 85.11 | 82.14 |
| TCoN (AAAI'20) | ResNet-101 | 87.22 | 89.14 | 88.18 |
| MA$^2$L-TD (WACV'22) | ResNet-101 | 85.00 | 86.59 | 85.80 |
| Source-only ($\mathcal{S}_{\text{only}}$) | I3D | 80.27 | 88.79 | 84.53 |
| DANN (JMLR'16) | I3D | 80.83 | 88.09 | 84.46 |
| ADDA (CVPR'17) | I3D | 79.17 | 88.44 | 83.81 |
| TA$^3$N (ICCV'19) | I3D | 81.38 | 90.54 | 85.96 |
| SAVA (ECCV'20) | I3D | 82.22 | 91.24 | 86.73 |
| CoMix (NeurIPS'21) | I3D | 86.66 | 93.87 | 90.22 |
| CO$^2$A (WACV'22) | I3D | 87.78 | 95.79 | 91.79 |
| **TranSVAE (Ours)** | **I3D** | **87.78** | **98.95** | **93.37** |
| Supervised-target ($\mathcal{T}_{\text{sup}}$) | I3D | 95.00 | 96.85 | 95.93 |

---

[3]The existing widely adopted backbones for video-based UDA include ResNet-101 and I3D. However, more recent backbones, *e.g.* Transformer-based or VideoMAE-based [36, 40], are promising to be used in this task.

Table 2: UDA performance comparisons to approaches using multi-modality data as the input.

| Task | $\mathcal{S}_{only}$ | MM-SADA | STCDA | CMCD | A3R | CleanAdapt | CycDA | MixDANN | CIA | **TranSVAE** |
|---|---|---|---|---|---|---|---|---|---|---|
| $U \rightarrow H$ | 86.1 | 84.2 | 83.1 | 84.7 | / | **89.8** | 88.1 | 82.2 | **88.3** | 87.8 (+1.7) |
| $H \rightarrow U$ | 92.5 | 91.1 | 92.1 | 92.8 | / | **99.2** | 98.0 | 92.8 | 94.1 | 99.0 (+6.5) |
| Average ↑ | 89.3 | 87.7 | 87.6 | 88.8 | / | **94.5** | 93.1 | 87.5 | 91.2 | 93.4 (+4.1) |
| $D_1 \rightarrow D_2$ | 43.2 | 49.5 | 52.0 | 50.3 | **53.2** | 52.7 | / | 56.0 | 52.5 | 50.5 (+7.3) |
| $D_1 \rightarrow D_3$ | 42.5 | 44.1 | 45.5 | 46.3 | **52.1** | 47.0 | / | 47.3 | 47.8 | 50.3 (+7.8) |
| $D_2 \rightarrow D_1$ | 43.0 | 48.2 | 49.0 | 49.5 | **51.9** | 46.2 | / | 50.3 | 49.8 | 50.3 (+7.3) |
| $D_2 \rightarrow D_3$ | 48.0 | 52.7 | 52.5 | 52.0 | 55.5 | 52.7 | / | 52.4 | 53.2 | **58.6** (+10.6) |
| $D_3 \rightarrow D_1$ | 43.0 | 50.9 | **52.6** | 48.7 | 51.5 | 47.8 | / | 51.0 | 52.2 | 48.0 (+5.0) |
| $D_3 \rightarrow D_2$ | 55.5 | 56.1 | 55.6 | 56.3 | **63.2** | 54.4 | / | 54.7 | 57.6 | 58.0 (+2.5) |
| Average ↑ | 45.9 | 50.3 | 51.2 | 51.0 | **54.1** | 50.3 | / | 52.0 | 52.2 | 52.6 (+6.7) |

**Competitors**. We compared methods from three lines. We first consider the *source-only* ($\mathcal{S}_{only}$) and *supervised-target* ($\mathcal{T}_{sup}$) which uses only labeled source data and only labeled target data, respectively. These two baselines serve as the lower and upper bounds for our tasks. Secondly, we consider five popular image-based UDA methods by simply ignoring temporal information, namely DANN [14], JAN [25], ADDA [38], AdaBN [21], and MCD [32]. Lastly and most importantly, we compare recent SoTA video-based UDA methods, including TA[3]N [7], SAVA [10], TCoN [29], ABG [26], CoMix [31], CO[2]A [37], and MA[2]L-TD [8]. All these methods use single modality features. We directly quote numbers reported in published papers whenever possible. There exist recent works conducting video-based UDA using multi-modal data, *e.g.* RGB + Flow. Although *TranSVAE* solely uses RGB features, we still take this set of methods into account. Specifically, we consider MM-SADA [28], STCDA [34], CMCD [18], A3R [48], CleanAdapt [12], CycDA [23], MixDANN [44] and CIA [43].

## 4.3 Comparative Study

**Results on UCF-HMDB**. Tab. 1 shows comparisons of *TranSVAE* with baselines and SoTA methods on UCF-HMDB. The best result among all the baselines is highlighted using bold. Overall, methods using the I3D backbone [6] achieve better results than those using ResNet-101. Our *TranSVAE* consistently outperforms all previous methods. In particular, *TranSVAE* achieves 93.37% average accuracy, improving the best competitor CO[2]A [37], with the same I3D backbone [6], by 1.38%. Surprisingly, we observe that *TranSVAE* even yields better results (by a 2.1% improvement) than the supervised-target ($\mathcal{T}_{sup}$) baseline. This is because the $H \rightarrow U$ task already has a good performance without adaptation, *i.e.*, 88.79% for the source-only ($\mathcal{S}_{only}$) baseline, and thus the target pseudo-labels used in *TranSVAE* are almost correct. By further aligning domains and equivalently augmenting training data, *TranSVAE* outperforms $\mathcal{T}_{sup}$ which is only trained with target data.

**Results on Jester & Epic-Kitchens**. Tab. 3 shows the comparison results on the Jester and Epic-Kitchens benchmarks. We can see that our *TranSVAE* is the clear winner among all the methods on all the tasks. Specifically, *TranSVAE* achieves a 1.4% improvement and a 9.4% average improvement over the runner-up baseline CoMix [31] on Jester and Epic-Kitchens, re-

Table 3: Comparison results on Jester and Epic-Kitchens.

| Task | $\mathcal{S}_{only}$ | DANN | ADDA | TA[3]N | CoMix | **TranSVAE** | $\mathcal{T}_{sup}$ |
|---|---|---|---|---|---|---|---|
| $J_{\mathcal{S}} \rightarrow J_{\mathcal{T}}$ | 51.5 | 55.4 | 52.3 | 55.5 | 64.7 | **66.1** (+14.6) | 95.6 |
| $D_1 \rightarrow D_2$ | 32.8 | 37.7 | 35.4 | 34.2 | 42.9 | **50.5** (+17.7) | 64.0 |
| $D_1 \rightarrow D_3$ | 34.1 | 36.6 | 34.9 | 37.4 | 40.9 | **50.3** (+16.2) | 63.7 |
| $D_2 \rightarrow D_1$ | 35.4 | 38.3 | 36.3 | 40.9 | 38.6 | **50.3** (+14.9) | 57.0 |
| $D_2 \rightarrow D_3$ | 39.1 | 41.9 | 40.8 | 42.8 | 45.2 | **58.6** (+19.5) | 63.7 |
| $D_3 \rightarrow D_1$ | 34.6 | 38.8 | 36.1 | 39.9 | 42.3 | **48.0** (+13.4) | 57.0 |
| $D_3 \rightarrow D_2$ | 35.8 | 42.1 | 41.4 | 44.2 | 49.2 | **58.0** (+22.2) | 64.0 |
| Average ↑ | 35.3 | 39.2 | 37.4 | 39.9 | 43.2 | **52.6** (+17.3) | 61.5 |

spectively. This verifies the superiority of *TranSVAE* over others in handling video-based UDA. However, we also notice that the accuracy gap between CoMix and $\mathcal{T}_{sup}$ is still significant on Jester. This is because the large-scale Jester dataset contains highly heterogeneous data across domains, *e.g.*, the source domain contains videos of the rolling hand forward, while the target domain only consists of videos of the rolling hand backward. This leaves much room for improvement in the future.

**Compare to Multi-Modal Methods**. We further compare with four recent video-based UDA methods that use multi-modalities, *e.g.* RGB features, and optical flows, although *TranSVAE* only uses RGB features. Surprisingly, *TranSVAE* achieves better average results than seven out of eight multi-modal methods and is worse than CleanAdapt [12] on UCF-HMDB and A3R [48] on Epic-Kitchens. Considering *TranSVAE* only uses single-modality data, we are confident that there exists great potential for further improvements of *TranSVAE* with multi-modal data taken into account.

Table 5: Loss separation study on different video-based UDA tasks by dropping each loss sequentially.

| $\mathcal{L}_{svae}$ | $\mathcal{L}_{cls}$ | $\mathcal{L}_{adv}$ | $\mathcal{L}_{mi}$ | $\mathcal{L}_{ctc}$ | PL | U $\rightarrow$ H | H $\rightarrow$ U | $J_S \rightarrow J_T$ | $D_1 \rightarrow D_2$ | $D_1 \rightarrow D_3$ | $D_2 \rightarrow D_1$ | $D_2 \rightarrow D_3$ | $D_3 \rightarrow D_1$ | $D_3 \rightarrow D_2$ | Avg. |
|---|---|---|---|---|---|---|---|---|---|---|---|---|---|---|---|
| ✓ | | ✓ | ✓ | ✓ | ✓ | 18.61 | 26.62 | 22.92 | 34.00 | 30.29 | 33.79 | 30.49 | 28.51 | 34.27 | 28.83 |
| ✓ | ✓ | | ✓ | ✓ | ✓ | 83.06 | 93.52 | 48.07 | 40.93 | 43.33 | 43.91 | 51.13 | 41.84 | 52.67 | 55.38 |
| ✓ | ✓ | ✓ | | ✓ | ✓ | 85.83 | 93.52 | 65.12 | 46.67 | 48.56 | 49.43 | 55.34 | 45.52 | 54.53 | 60.60 |
| ✓ | ✓ | ✓ | ✓ | | ✓ | 83.89 | 95.80 | 64.89 | 48.53 | 48.25 | 48.96 | 54.21 | 45.52 | 55.73 | 60.64 |
| ✓ | ✓ | ✓ | ✓ | ✓ | | 87.22 | 94.40 | 64.64 | 49.87 | 48.25 | 49.66 | 56.47 | 47.59 | 55.07 | 61.46 |
| ✓ | ✓ | ✓ | ✓ | ✓ | ✓ | **87.78** | **98.95** | **66.10** | **50.53** | **50.31** | **50.34** | **58.62** | **48.04** | **58.00** | **63.19** |

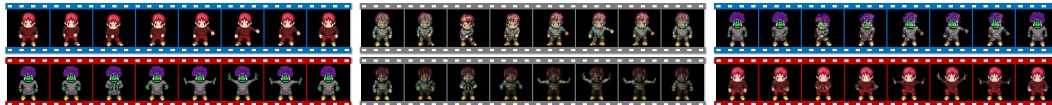

Figure 4: Domain disentanglement and transfer examples. **Left:** Video sequence inputs for $\mathcal{D} = \mathbf{P}_1$ ("Human", ■) and $\mathcal{D} = \mathbf{P}_2$ ("Alien", ■). **Middle:** Reconstructed sequences (■) with $\mathbf{z}_1^{\mathcal{D}}, ..., \mathbf{z}_T^{\mathcal{D}}$. **Right:** Domain transferred sequences with exchanged $\mathbf{z}_d^{\mathcal{D}}$.

## 4.4 Property Analysis

**Disentanglement Analysis**. We analyze the disentanglement effect of *TranSVAE* on Sprites [22] and show results in Fig. 4. The left subfigure shows the original sequences of the two domains. The "Human" and "Alien" are completely different appearances and the former is *casting spells* while the latter is *slashing*. The middle subfigure shows the sequences reconstructed only using $\{\mathbf{z}_1^{\mathcal{D}}, ..., \mathbf{z}_T^{\mathcal{D}}\}$. It can be clearly seen that the two sequences keep the same action as the corresponding original ones. However, if we only focus on the appearance characteristics, it is difficult to distinguish the domain to which the sequences belong. This indicates that $\{\mathbf{z}_1^{\mathcal{D}}, ..., \mathbf{z}_T^{\mathcal{D}}\}$ are indeed *domain-invariant* and well encode the semantic information. The right subfigure shows the sequences reconstructed by exchanging $\mathbf{z}_d^{\mathcal{D}}$, which results in two sequences with the same actions but exchanged appearance. This verifies that $\mathbf{z}_d^{\mathcal{D}}$ is representing the appearance information, which is actually the *domain-related* information in this example. This property study sufficiently supports that *TranSVAE* can successfully disentangle the domain information from other information, with the former embedded in $\mathbf{z}_d^{\mathcal{D}}$ and the latter embedded in $\{\mathbf{z}_1^{\mathcal{D}}, ..., \mathbf{z}_T^{\mathcal{D}}\}$.

**Complexity Analysis**. We further conduct complexity analysis on our *TranSVAE*. Specifically, we compare the number of trainable parameters, multiply-accumulate operations (MACs), floating-point operations (FLOPs), and inference frame-per-second (FPS) with existing baselines including TA[3]N [7], CO[2]A [37], and CoMix

Table 4: Comparison results on model complexity.

| Methods | Trainable Params | MACs | FLOPs | FPS |
|---|---|---|---|---|
| TA[3]N | 7.6880 M | 18.2318 G | 36.4636 G | 0.0134 s |
| CoMix | 30.3688 M | 18.5640 G | 37.1280 G | 0.0157 s |
| CO[2]A | 23.6720 M | 18.1884 G | 36.3768 G | 0.0127 s |
| **TranSVAE** | 12.7419 M | 18.2657 G | 36.5314 G | 0.0133 s |

[31]. All the comparison results are shown in Tab. 4. From the table, we observe that *TranSVAE* requires less trainable parameters than CO[2]A and CoMix. Although more trainable parameters are used than TA[3]N, *TranSVAE* achieves significant adaptation performance improvement than TA[3]N (see Tab. 1 and Tab. 3). Moreover, the MACs, FLOPs, and FPS are competitive among different methods. This is reasonable since all these approaches adopt the same I3D backbone.

**Ablation Study**. We now analyze the effectiveness of each loss term in Eq. (14). We compare with four variants of *TranSVAE*, each removing one loss term by equivalently setting the weight $\lambda$ to 0. The ablation results on UCF-HMDB, Jester, and Epic-Kitchens are shown in Tab. 5. As can be seen, removing $\mathcal{L}_{cls}$ significantly reduces the transfer performance in all the tasks. This is reasonable as $\mathcal{L}_{cls}$ is used to discover the discriminative features. Removing any of $\mathcal{L}_{adv}$, $\mathcal{L}_{mi}$, and $\mathcal{L}_{ctc}$ leads to an inferior result than the full *TranSVAE* setup, and removing $\mathcal{L}_{adv}$ is the most influential. This is because $\mathcal{L}_{adv}$ is used to explicitly reduce the temporal domain gaps. All these results demonstrate that every proposed loss matters in our framework.

We further conduct another ablation study by sequentially integrating $\mathcal{L}_{cls}$, $\mathcal{L}_{adv}$, $\mathcal{L}_{mi}$, and $\mathcal{L}_{ctc}$ into our sequential VAE structure using UCF-HMDB. We use this integration order based on the average positive improvement that a loss brings to *TranSVAE* as shown in Tab. 5. We also take advantage of t-SNE [39] to visualize the features learned by these different variants. We plot two sets of t-SNE

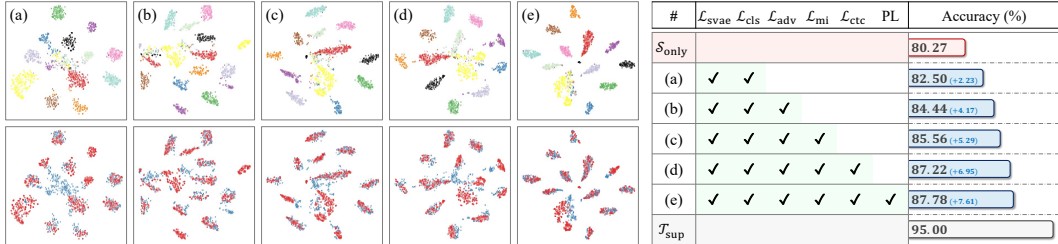

Figure 5: Loss integration studies on **U → H**. **Left:** The t-SNE plots for class-wise (top row) and domain (bottom row, red source & blue target) features. **Right:** Ablation results (%) by adding each loss sequentially, *i.e.*, row (a) - row (e).

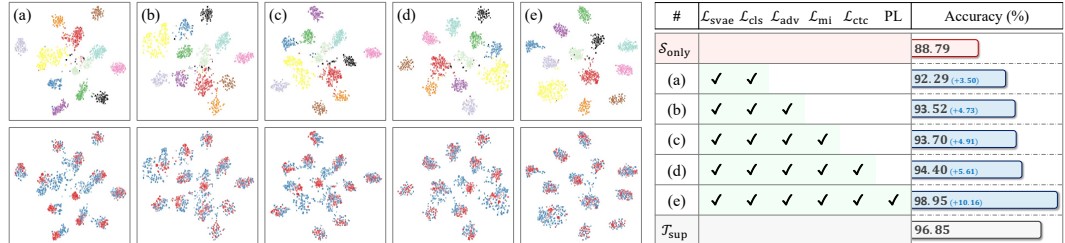

Figure 6: Loss integration studies on **H → U**. **Left:** The t-SNE plots for class-wise (top row) and domain (bottom row, red source & blue target) features. **Right:** Ablation results (%) by adding each loss sequentially, *i.e.*, row (a) - row (e).

figures, one using the class-wise label and another using the domain label. Fig. 5 and Fig. 6 show the visualization and the quantitative results. As can be seen from the t-SNE feature visualizations, adding a new component improves both the domain and semantic alignments, and the best alignment is achieved when all the components are considered. The quantitative results further show that the transfer performance gradually increases with the sequential integration of the four components, which again verifies the effectiveness of each component in *TranSVAE*. More ablation study results can be found in the Appendix.

## 5   Conclusion and Limitation

In this paper, we proposed a *TranSVAE* framework for video-based UDA tasks. Our key idea is to explicitly disentangle the domain information from other information during the adaptation. We developed a novel sequential VAE structure with two sets of latent factors and proposed four constraints to regulate these factors for adaptation purposes. Note that disentanglement and adaptation are interactive and complementary. All the constraints serve to achieve a good disentanglement effect with the two-level domain divergence minimization. Extensive empirical studies clearly verify that *TranSVAE* consistently offers performance improvements compared with existing SoTA video-based UDA methods. We also find that *TranSVAE* outperforms those multi-modal UDA methods, although it only uses single-modality data. Comprehensive property analysis further shows that *TranSVAE* is an effective and promising method for video-based UDA.

We further discuss the limitations of the *TranSVAE* framework. These are also promising future directions. Firstly, the empirical evaluation is mainly on the action recognition task. The performance of *TranSVAE* on other video-related tasks, *e.g.* video segmentation, is not tested. Secondly, *TranSVAE* is only evaluated on the typical two-domain transfer scenario. The multi-source transfer case is not considered but is worthy of further study. Thirdly, although *TranSVAE* exhibits better performance than multi-modal transfer methods, its current version does not consider multi-modal data functionally and structurally. An improved *TranSVAE* with the capacity of using multi-modal data is expected to further boost the adaptation performance. Fourthly, current empirical evaluations are mainly based on the I3D backbone, more advanced backbones, *e.g.*, [36, 40], are expected to be explored for further improvement. Lastly, *TranSVAE* handles the spatial divergence by disentangling the static domain-specific latent factors out. However, it may happen that spatial divergence is not completely captured by the static latent factors due to insufficient disentanglement.

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

# Appendix

In this appendix, we supplement the following materials to support the findings and conclusions drawn in the main body of this paper:

- Sec. A provides additional implementation details to assist reproductions.
- Sec. B contains additional comparative and ablation results of the *TranSVAE* framework.
- Sec. C elaborates on the broader impact of this work.
- Sec. D acknowledges the public resources used during the course of this work.

# A  Additional Implementation Details

In this section, we provide more implementation details including the I3D feature extraction procedure, the concrete model architecture, and the hyperparameter selection in our proposed *TranSVAE* framework. We also provide detailed instructions for our live demo.

## A.1  Notations

We present Tab. A to summarize the notations used in this paper.

## A.2  I3D Feature Extractions

We extract the I3D RGB features following the routine described in SAVA [10]. Given a video sequence, 16 frames along clips are sampled by sliding a temporal window with a temporal stride of 1. Specifically, for each frame in the video, the temporal window consists of its previous seven frames and the following eight frames. Zero padding is used for the beginning and the end of the video. We then feed the sliding windows to the I3D backbone to extract features, which results in a 1024-dimensional feature vector for each frame of the video.

## A.3  Model Architecture

We now provide the detailed model architecture of our *TranSVAE*. In Fig. A, we show the model with the *image* as the input, where the encoder and decoder are more complex convolutional and deconvolutional layers. For the model with the RGB *features* as the input, we can simply replace the encoder and decoder with fully connected linear layers. Note that the dimensionality of all the modules shown above is uniformly applied in all the experiments.

## A.4  Hyperparameter Selection

There are several hyperparameters used in *TranSVAE*, including the balancing weights $\lambda_1$, $\lambda_2$, $\lambda_3$, $\lambda_4$, the number of the video frames $T$, and the confidence threshold $\eta$ for generating target pseudo-labels. For $\lambda_1$ to $\lambda_4$, we select from the value set $\{1e^{-3}, 1e^{-2}, 1e^{-1}, 0.5, 1, 5, 10, 50, 100, 1000\}$. For $T$, we select from $\{5, 6, 7, 8, 9, 10\}$. For $\eta$, we set its value range from 0.9 to 1.0 with a step of 0.01.

We set a high-value range of $\eta$ to ensure a high confidence score on the correctness of the target pseudo-labels. Following the common protocol used in video-based UDA, we conduct an extensive grid search regarding these hyperparameters on the validation set of each transfer task. Tab. B summarizes the exact used values of these hyperparameters for the UCF-HMDB [35, 20], Jester [27], and Epic-Kitchens [11] UDA benchmarks. For the Sprites [22] dataset, we do not do a hyperparameter search as the data is quite simple. We simply set $T$ as 8, which is the original length of the video sequence. The confidence threshold is set to be 0.99, and $\lambda_1$ to $\lambda_4$ are all set to be 1.

## A.5  Demo Instruction

As mentioned in the main body, we include a live demo for our *TranSVAE* framework. This demo can be accessed at: https://huggingface.co/spaces/ldkong/TranSVAE. Here we include the detailed instructions for playing with this demo.

Table A: Summary of notations used in this paper.

| Notation | Description |
|---|---|
| $\mathcal{D}$ | Domain |
| $\mathcal{S}/\mathcal{T}$ | Source domain / Target domain |
| $\mathbf{V}^{\mathcal{D}}$ | A video sequence from domain $\mathcal{D}$ |
| $\mathbf{V}_i^{\mathcal{D}}, y_i^{\mathcal{D}}$ | The $i$-th video sequence and the corresponding action label of domain $\mathcal{D}$ |
| $\mathbf{x}_i^{\mathcal{D}}, ..., \mathbf{x}_T^{\mathcal{D}}$ | $T$ frames of images in the video sequence $\mathbf{V}^{\mathcal{D}}$ |
| $\mathbf{z}_i^{\mathcal{D}}, ..., \mathbf{z}_T^{\mathcal{D}}$ | The dynamic latent factors of the video sequence from domain $\mathcal{D}$ |
| $\mathbf{z}_d^{\mathcal{D}}$ | The static latent factors of the video sequence from domain $\mathcal{D}$ |
| $\mathbf{x}_{<t}^{\mathcal{D}}$ | The input sequence before timestamp $t$ |
| $\mathbf{x}_{1:T}^{\mathcal{D}}$ | The full input sequence from $t=1$ to $t=T$. |

Table B: Summary of the best-possible hyperparameter values of the *TranSVAE* framework for each UDA task in our experiments after an extensive grid search.

| Task | Backbone | $T$ | $\eta$ | $\lambda_1$ | $\lambda_2$ | $\lambda_3$ | $\lambda_4$ |
|---|---|---|---|---|---|---|---|
| $\mathbf{U} \rightarrow \mathbf{H}$ | I3D | 8 | 0.93 | 50 | 1 | 1 | 1 |
| $\mathbf{H} \rightarrow \mathbf{U}$ | I3D | 9 | 0.96 | 0.5 | 0.1 | 10 | 1 |
| $\mathbf{J}_{\mathcal{S}} \rightarrow \mathbf{J}_{\mathcal{T}}$ | I3D | 6 | 0.95 | 0.001 | 10 | 100 | 10 |
| $\mathbf{D}_1 \rightarrow \mathbf{D}_2$ | I3D | 9 | 0.96 | 50 | 10 | 5 | 100 |
| $\mathbf{D}_1 \rightarrow \mathbf{D}_3$ | I3D | 10 | 1 | 1 | 0.5 | 0.1 | 100 |
| $\mathbf{D}_2 \rightarrow \mathbf{D}_1$ | I3D | 8 | 0.93 | 100 | 10 | 5 | 100 |
| $\mathbf{D}_2 \rightarrow \mathbf{D}_3$ | I3D | 10 | 0.91 | 0.5 | 10 | 50 | 100 |
| $\mathbf{D}_3 \rightarrow \mathbf{D}_1$ | I3D | 8 | 0.91 | 100 | 10 | 50 | 100 |
| $\mathbf{D}_3 \rightarrow \mathbf{D}_2$ | I3D | 9 | 0.91 | 1000 | 1 | 0.1 | 100 |

This demo is built upon Hugging Face Spaces[4], which provides concise and easy-to-use live demo interfaces. Our demo consists of one input interface and one output interface as shown in Fig. B. Specifically, the appearances of the Sprites avatars are fully controlled by four attributes, *i.e.*, body, hair color, top wear, and bottom wear. We construct two domains, $\mathbf{P}_1$ and $\mathbf{P}_2$. $\mathbf{P}_1$ uses the "Human" body while $\mathbf{P}_1$ uses the "Alien" body. The attribute pools of "Human" and "Alien" are non-overlapping across domains, resulting in completely heterogeneous $\mathbf{P}_1$ and $\mathbf{P}_2$. Each video sequence contains eight frames in total.

For conducting domain disentanglement and transfer with *TranSVAE*, users are free to choose the action and the appearance of the avatars on the left-hand side of the interface. Next, simply click the "Submit" button and the adaptation results will display on the right-hand side of the interface in a few seconds. The outputs include:

- The **1st** column: The original input of the "Human" and "Alien" avatars, *i.e.*, $\{\mathbf{x}_1^{\mathbf{P}_1}, ..., \mathbf{x}_8^{\mathbf{P}_1}\}$ and $\{\mathbf{x}_1^{\mathbf{P}_2}, ..., \mathbf{x}_8^{\mathbf{P}_2}\}$;

- The **2nd** column: The reconstructed "Human" and "Alien" avatars $\{\widetilde{\mathbf{x}}_1^{\mathbf{P}_1}, ..., \widetilde{\mathbf{x}}_8^{\mathbf{P}_1}\}$ and $\{\widetilde{\mathbf{x}}_1^{\mathbf{P}_2}, ..., \widetilde{\mathbf{x}}_8^{\mathbf{P}_2}\}$;

- The **3rd** column: The reconstructed "Human" and "Alien" avatars using only $\{\mathbf{z}_1^{\mathcal{D}}, ..., \mathbf{z}_8^{\mathcal{D}}\}$, $\mathcal{D} \in \{\mathbf{P}_1, \mathbf{P}_2\}$, which are domain-invariant;

---

[4] https://huggingface.co

TranSVAE(
  (**encoder**): encoder(
    (c1): dcgan_conv(
      (main): Sequential(
        (0): Conv2d(3, 64, kernel_size=(4, 4), stride=(2, 2), padding=(1, 1))
        (1): BatchNorm2d(64, eps=1e-05, momentum=0.1, affine=True, track_running_stats=True)
        (2): LeakyReLU(negative_slope=0.2, inplace=True) ) )
    (c2): dcgan_conv( (main): Sequential(
        (0): Conv2d(64, 128, kernel_size=(4, 4), stride=(2, 2), padding=(1, 1))
        (1): BatchNorm2d(128, eps=1e-05, momentum=0.1, affine=True, track_running_stats=True)
        (2): LeakyReLU(negative_slope=0.2, inplace=True) ) )
    (c3): dcgan_conv( (main): Sequential(
        (0): Conv2d(128, 256, kernel_size=(4, 4), stride=(2, 2), padding=(1, 1))
        (1): BatchNorm2d(256, eps=1e-05, momentum=0.1, affine=True, track_running_stats=True)
        (2): LeakyReLU(negative_slope=0.2, inplace=True) ) )
    (c4): dcgan_conv( (main): Sequential(
        (0): Conv2d(256, 512, kernel_size=(4, 4), stride=(2, 2), padding=(1, 1))
        (1): BatchNorm2d(512, eps=1e-05, momentum=0.1, affine=True, track_running_stats=True)
        (2): LeakyReLU(negative_slope=0.2, inplace=True) ) )
    (c5): Sequential(
        (0): Conv2d(512, 1024, kernel_size=(4, 4), stride=(1, 1))
        (1): BatchNorm2d(1024, eps=1e-05, momentum=0.1, affine=True, track_running_stats=True)
        (2): Tanh() )
  )
  (**decoder**): decoder_woSkip
    (upc1): Sequential(
        (0): ConvTranspose2d(1024, 512, kernel_size=(4, 4), stride=(1, 1))
        (1): BatchNorm2d(512, eps=1e-05, momentum=0.1, affine=True, track_running_stats=True)
        (2): LeakyReLU(negative_slope=0.2, inplace=True) )
    (upc2): dcgan_upconv( (main): Sequential(
        (0): ConvTranspose2d(512, 256, kernel_size=(4, 4), stride=(2, 2), padding=(1, 1))
        (1): BatchNorm2d(256, eps=1e-05, momentum=0.1, affine=True, track_running_stats=True)
        (2): LeakyReLU(negative_slope=0.2, inplace=True) ) )
    (upc3): dcgan_upconv( (main): Sequential(
        (0): ConvTranspose2d(256, 128, kernel_size=(4, 4), stride=(2, 2), padding=(1, 1))
        (1): BatchNorm2d(128, eps=1e-05, momentum=0.1, affine=True, track_running_stats=True)
        (2): LeakyReLU(negative_slope=0.2, inplace=True) ) )
    (upc4): dcgan_upconv( (main): Sequential(
        (0): ConvTranspose2d(128, 64, kernel_size=(4, 4), stride=(2, 2), padding=(1, 1))
        (1): BatchNorm2d(64, eps=1e-05, momentum=0.1, affine=True, track_running_stats=True)
        (2): LeakyReLU(negative_slope=0.2, inplace=True) ) )
    (upc5): Sequential(
        (0): ConvTranspose2d(64, 3, kernel_size=(4, 4), stride=(2, 2), padding=(1, 1))
        (1): Sigmoid() )
  )
  (**relu**): LeakyReLU(negative_slope=0.1)
  (**dropout_f**): Dropout(p=0.5, inplace=False)
  (**dropout_v**): Dropout(p=0.5, inplace=False)
  (**z_prior_lstm_ly1**): LSTMCell(512, 512)
  (**z_prior_lstm_ly2**): LSTMCell(512, 512)
  (**z_prior_mean**): Linear(in_features=512, out_features=512, bias=True)
  (**z_prior_logvar**): Linear(in_features=512, out_features=512, bias=True)
  (**z_lstm**): LSTM(1024, 512, batch_first=True, bidirectional=True)
  (**f_mean**): Linear(in_features=1024, out_features=512, bias=True)
  (**f_logvar**): Linear(in_features=1024, out_features=512, bias=True)
  (**z_rnn**): RNN(1024, 512, batch_first=True)
  (**z_mean**): Linear(in_features=512, out_features=512, bias=True)
  (**z_logvar**): Linear(in_features=512, out_features=512, bias=True)
  (**fc_feature_domain_frame**): Linear(in_features=512, out_features=512, bias=True)
  (**fc_classifier_domain_frame**): Linear(in_features=512, out_features=2, bias=True)
  (**TRN**)
  (**bn_trn_S**): BatchNorm1d(256, eps=1e-05, momentum=0.1, affine=True, track_running_stats=True)
  (**bn_trn_T**): BatchNorm1d(256, eps=1e-05, momentum=0.1, affine=True, track_running_stats=True)
  (**fc_feature_domain_video**): Linear(in_features=256, out_features=256, bias=True)
  (**fc_classifier_domain_video**): Linear(in_features=256, out_features=2, bias=True)
  (**relation_domain_classifier_all**): ModuleList(
    (0): Sequential(
        (0): Linear(in_features=256, out_features=256, bias=True)
        (1): ReLU()
        (2): Linear(in_features=256, out_features=2, bias=True) )
    (1): Sequential(
        (0): Linear(in_features=256, out_features=256, bias=True)
        (1): ReLU()
        (2): Linear(in_features=256, out_features=2, bias=True) )
    (2): Sequential(
        (0): Linear(in_features=256, out_features=256, bias=True)
        (1): ReLU()
        (2): Linear(in_features=256, out_features=2, bias=True) )
    (3): Sequential(
        (0): Linear(in_features=256, out_features=256, bias=True)
        (1): ReLU()
        (2): Linear(in_features=256, out_features=2, bias=True) )
    (4): Sequential(
        (0): Linear(in_features=256, out_features=256, bias=True)
        (1): ReLU()
        (2): Linear(in_features=256, out_features=2, bias=True) )
    (5): Sequential(
        (0): Linear(in_features=256, out_features=256, bias=True)
        (1): ReLU()
        (2): Linear(in_features=256, out_features=2, bias=True) )
    (6): Sequential(
        (0): Linear(in_features=256, out_features=256, bias=True)
        (1): ReLU()
        (2): Linear(in_features=256, out_features=2, bias=True) )
  )
  (**pred_classifier_video**): Linear(in_features=256, out_features=15, bias=True)
  (**fc_feature_domain_latent**): Linear(in_features=512, out_features=512, bias=True)
  (**fc_classifier_doamin_latent**): Linear(in_features=512, out_features=2, bias=True)
)

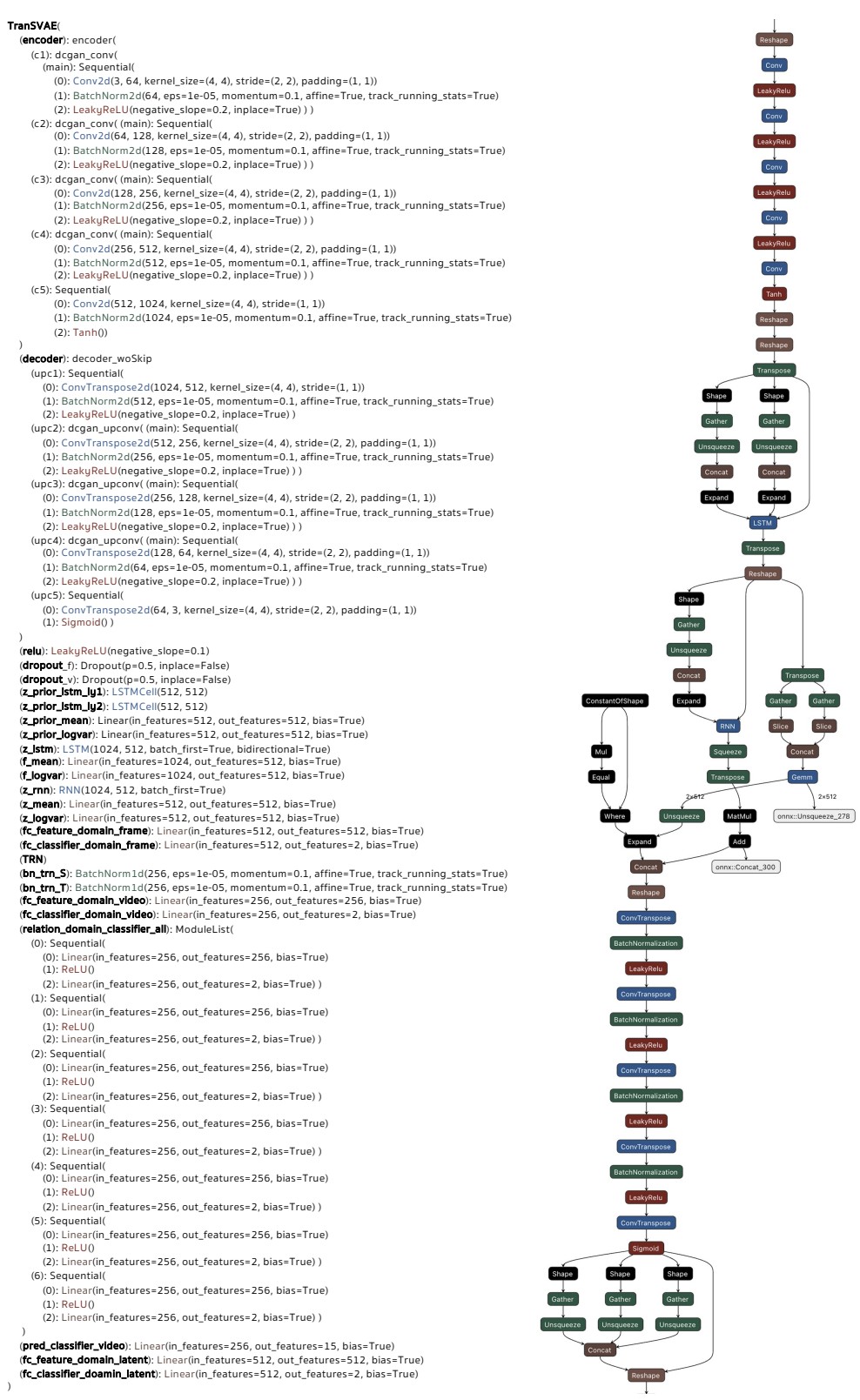

Figure A: The neural network structure (left) and a Netron graph (right) of the proposed *TranSVAE* framework. Zoom-ed in for the details.

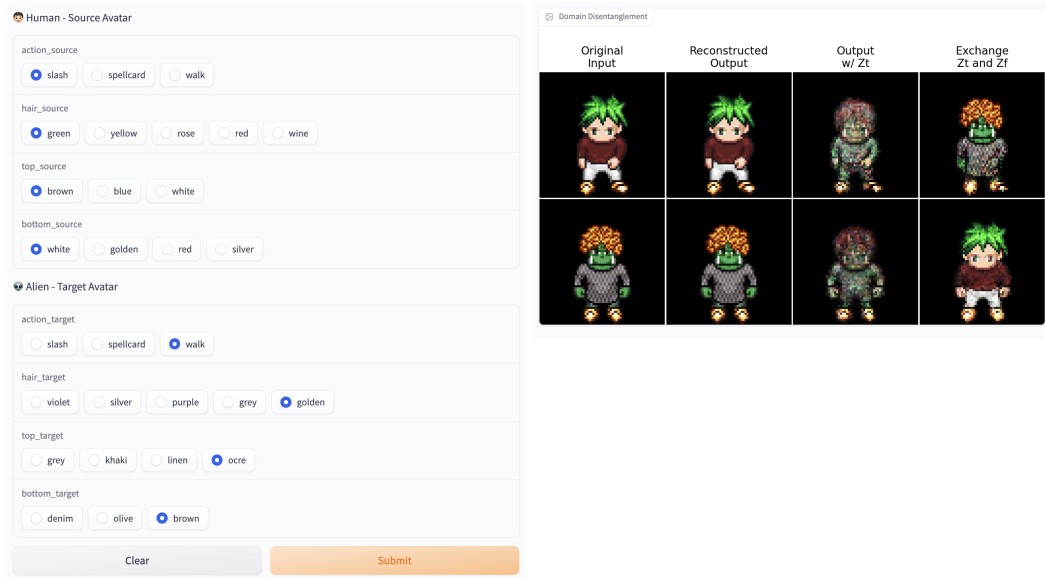

Figure B: The input (left) and output (right) interfaces of our live demo. Users are free to customize the actions and appearances of the source and target inputs (*i.e.*, the "Human" and "Alien" avatars) in the left-hand side and use them for disentanglement and transfer as shown in the right-hand side.

Table C: Ablation results without and with disentanglement term.

| $\mathcal{L}_{\text{svae}}$ | $\mathcal{L}_{\text{cls}}$ | $\mathcal{L}_{\text{adv}}$ | $\mathcal{L}_{\text{mi}}$ | $\mathcal{L}_{\text{ctc}}$ | $\mathbf{U} \to \mathbf{H}$ | $\mathbf{H} \to \mathbf{U}$ |
|---|---|---|---|---|---|---|
| | ✓ | ✓ | | | 81.67 | 90.54 |
| | ✓ | | ✓ | | 79.44 | 90.37 |
| | ✓ | | | ✓ | 81.11 | 90.37 |
| | ✓ | ✓ | ✓ | ✓ | 82.22 | 91.07 |
| ✓ | ✓ | ✓ | | | 84.44 | 93.52 |
| ✓ | ✓ | | ✓ | | 83.06 | 93.70 |
| ✓ | ✓ | | | ✓ | 83.61 | 91.42 |
| ✓ | ✓ | ✓ | ✓ | ✓ | **87.78** | **98.95** |

- The **4th** column: The reconstructed "Human" and "Alien" avatars by exchanging $\mathbf{z}_d^{\mathcal{D}}$, which results in two sequences with the same actions but exchanged appearance, *i.e.*, domain disentanglement and transfer.

# B  Additional Experimental Results

In this section, we provide additional quantitative and qualitative results for our *TranSVAE* framework.

## B.1  Additional Ablation Studies

**Component Integration Study**. We further analyze the effect of each loss term without disentanglement, *i.e.*, not adding $\mathcal{L}_{\text{svae}}$, on the UCF-HMDB dataset, as shown in Tab. C. Without $\mathcal{L}_{\text{svae}}$, the framework is to learn representations with corresponding constraints as many existing video-based UDA methods do. Precisely, we have the following remarks for the top-half table:

- Only with $\mathcal{L}_{\text{adv}}$, the framework is equivalent to TA$^3$N [7], which learns the domain-invariant temporal representation. We can see the adaptation results are similar to that of [7].
- $\mathcal{L}_{\text{mi}}$ is a term specifically designed for disentanglement purposes. Thus, the effect of using $\mathcal{L}_{\text{mi}}$ alone for adaptation is random as verified in the above table, achieving worse results

Table D: Ablation results for the frame number $T$.

| Task | 5 | 6 | 7 | 8 | 9 | 10 | 12 | 14 | 16 | 20 |
|------|------|------|------|------|------|------|------|------|------|------|
| $\mathbf{U} \to \mathbf{H}$ | 84.44 | 86.11 | 84.17 | **87.78** | 84.72 | 85.28 | 83.89 | 84.44 | 82.78 | 85.00 |
| $\mathbf{H} \to \mathbf{U}$ | 96.85 | 94.22 | 98.42 | 94.75 | **98.95** | 93.70 | 93.87 | 94.40 | 94.05 | 94.40 |

Table E: Ablation results for the pseudo-label threshold $\eta$.

| Task | *w/o* | 0.90 | 0.91 | 0.92 | 0.93 | 0.94 | 0.95 | 0.96 | 0.97 | 0.98 | 0.99 |
|------|------|------|------|------|------|------|------|------|------|------|------|
| $\mathbf{U} \to \mathbf{H}$ | 87.22 (−0.56) | 86.94 | 87.22 | 87.50 | **87.78** | 86.11 | 86.67 | 86.94 | 86.39 | 86.11 | 86.39 |
| $\mathbf{H} \to \mathbf{U}$ | 94.40 (−4.55) | 98.42 | 97.37 | 98.77 | 98.60 | 98.25 | 98.25 | **98.95** | 97.90 | 97.72 | 95.27 |

       in the $\mathbf{U} \to \mathbf{H}$ task and better results in the $\mathbf{H} \to \mathbf{U}$ tasks compared with the source-only baseline.

- $\mathcal{L}_{\text{ctc}}$ enforces the learned static representation to be domain-specific and brings indirect benefit to adaptation, thus obtaining slightly better results than the source-only baseline.

In sum, without $\mathcal{L}_{\text{svae}}$, the adaptation performance of each loss term alone is far from optimal. Moreover, combining all the terms can slightly improve the adaptation performance over the individual ones, however, is still obviously worse than *TranSVAE* results. The above ablation study shows the necessity of the disentanglement in the *TranSVAE* framework and proves that $\mathcal{L}_{\text{svae}}$ is an indispensable loss term.

Regarding the bottom half table showing the results with $\mathcal{L}_{\text{adv}}$, we have the following remarks:

- Each individual term with $\mathcal{L}_{\text{adv}}$ achieves better adaptation results than the source-only baseline.

- Each individual term with $\mathcal{L}_{\text{adv}}$ consistently outperforms the corresponding ones without $\mathcal{L}_{\text{adv}}$.

- The final *TranSVAE* framework combined all the loss terms yields much better results than each individual case.

This set of ablation studies indicates that 1) each term in *TranSVAE* brings positive effects to adaptation, and 2) disentanglement and adaptation are interactive and complementary in our framework to obtain the best possible UDA results.

**Number of Frames** $T$. Tab. D shows the transfer performance with the variation of the number of frames $T$ on the UCF-HMDB dataset. As can be seen, the two tasks achieve the optimal performance with different $T$, specifically $T = 8$ for $\mathbf{U} \to \mathbf{H}$ and $T = 9$ for $\mathbf{H} \to \mathbf{U}$. Based on this observation, we apply a grid search on the validation set to obtain the optimal $T$ for each task in our experiments.

**Target Pseudo-Label Threshold** $\eta$. We show the sensitivity analyses of the transfer performance with respect to the target pseudo label threshold $\eta$ on the UCF-HMDB dataset in Tab. E. The results show that different tasks yield the best transfer result with different $\eta$, specifically $\eta = 0.93$ for $\mathbf{U} \to \mathbf{H}$ and $\eta = 0.96$ for $\mathbf{H} \to \mathbf{U}$. Thus, we also apply the grid search on the validation set to obtain the optimal $\eta$ for each task.

## B.2 Additional Qualitative Results

For those who cannot access our live demo, we have included more qualitative examples for domain disentanglement and transfer in Fig. C. We also provide a link[5] for GIFs demonstrating various disentanglement and reconstruction results.

In the GIFs link, we show two cases, the first two columns for the first one and the last two columns for the second one. Each case contains a source and a target cartoon character performing an action. For each row, we have the following remark:

---

[5]https://github.com/ldkong1205/TranSVAE#ablation-study

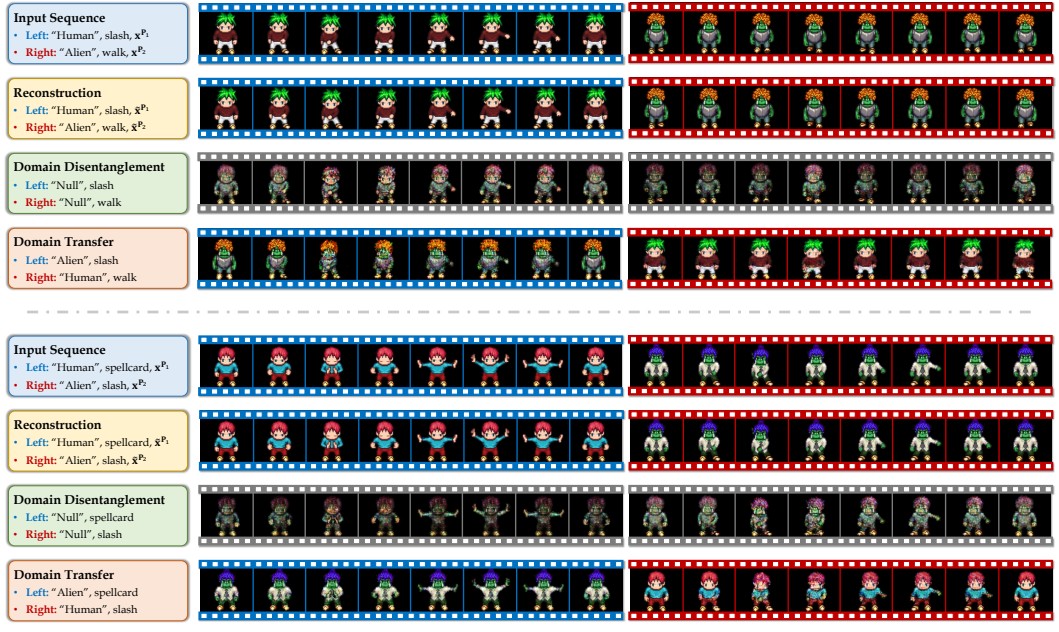

Figure C: Additional qualitative results for illustrating the domain disentanglement and transfer properties in our *TranSVAE* framework.

- The second row shows the reconstructed results of *TranSVAE*, *i.e.* $\mathbf{z}_d^{\mathcal{D}}$ and $\mathbf{z}_t^{\mathcal{D}}$. As can be seen, *TranSVAE* reconstructs the image with a high quality.

- The third row shows the reconstructed results only using the static latent factors, *i.e.*, $\mathbf{z}_d^{\mathcal{D}}$ and $\mathbf{0}_t^{\mathcal{D}}$, where we replace $\mathbf{z}_t^{\mathcal{D}}$ with zero vectors. As can be seen, the reconstructed results are basically static containing the appearance of the character which is the main domain difference in the Sprites dataset [22]. Specifically, we find they generally lack arms. This is reasonable as the target action is slashing or spelling cards with arms moving, and such dynamic information on the arm is captured by $\mathbf{z}_t^{\mathcal{D}}$.

- The fourth row shows the reconstructed results only using the dynamic latent factors, *i.e.*, $\mathbf{0}_d^{\mathcal{D}}$ and $\mathbf{z}_t^{\mathcal{D}}$, where we replace $\mathbf{z}_d^{\mathcal{D}}$ with zero vectors. As can be seen, the reconstructed results are performing the right action but the appearance is mixed up. This shows that $\mathbf{z}_t^{\mathcal{D}}$ are indeed domain-invariant and contain the semantic information.

- The last row shows the reconstructed results of exchanging the dynamic latent factors between domains, *i.e.*, $(\mathbf{0}_d^{S}, \mathbf{z}_t^{\mathcal{T}})$ and $(\mathbf{0}_d^{\mathcal{T}}, \mathbf{z}_t^{S})$. As can be seen, the reconstructed results are with the original appearance but perform the transferred action. This indicates the potential of *TranSVAE* for some style-transfer tasks.

## C  Broader Impact

This paper provides a novel transfer method to use cross-domain video data, which effectively helps reduce the annotation efforts in related video applications. Although the main empirical evaluation is on the video action recognition task, the model structure proposed in this paper is also applicable to other video-related tasks, such as action segmentation, video semantic segmentation, *etc*. More generally, the idea of disentangling domain information sheds light on other data modality style transfer tasks, *e.g.*, voice conversion. The negative impacts of this work are difficult to predict. However, as a deep model, our method shares some common pitfalls of the standard deep learning models, *e.g.*, demand for powerful computing resources, and vulnerability to adversarial attacks.

## D  Public Resources Used

We acknowledge the use of the following public resources, during the course of this work:

- $UCF_{101}$ [6] ............................................................................ Unknown
- $HMDB_{51}$ [7] .......................................................................... CC BY 4.0
- Jester [8] .............................................................................. Unknown
- Epic-Kitchens [9] .................................................................. CC BY-NC 4.0
- Sprites [10] ......................................................................... CC-BY-SA-3.0
- I3D [11] ........................................................................ Apache License 2.0
- TRN [12] ....................................................................... BSD 2-Clause License
- CoMix [13] .................................................................... Apache License 2.0
- $TA^3N$ [14] ........................................................................... MIT License
- $CO^2A$ [15] ............................................................................. Unknown
- PyTorch [16] ............................................................... PyTorch Custom License
- Netron [17] ........................................................................... MIT License
- flops-counter.pytorch [18] ........................................................... MIT License

---

[6] https://www.crcv.ucf.edu/data/UCF101.php
[7] https://serre-lab.clps.brown.edu/resource/hmdb-a-large-human-motion-database
[8] https://20bn.com/datasets/jester
[9] https://epic-kitchens.github.io/2021
[10] https://github.com/YingzhenLi/Sprites
[11] https://github.com/piergiaj/pytorch-i3d
[12] https://github.com/zhoubolei/TRN-pytorch
[13] https://github.com/CVIR/CoMix
[14] https://github.com/cmhungsteve/TA3N
[15] https://github.com/vturrisi/CO2A
[16] https://github.com/pytorch/pytorch
[17] https://github.com/lutzroeder/netron
[18] https://github.com/sovrasov/flops-counter.pytorch

