# OpenReview forum: "Unsupervised Video Domain Adaptation for Action Recognition: A Disentanglement Perspective"
_NeurIPS.cc/2023/Conference — NeurIPS 2023 poster_

### Official Review · Reviewer_qCdJ · 2023-07-05

**Soundness:** 2 fair
**Presentation:** 2 fair
**Contribution:** 3 good
**Rating:** 5
**Confidence:** 4

**Summary:**

The proposed method in this paper handles the spatial and temporal domain divergence separately via disentanglement.

Two sets of latent factors are considered (i) static information encoding and (ii) dynamic information encoding.

The transfer sequential VAE framework is proposed to model the generation of cross-domain videos and better serve for the adaptation process.

The proposed method facilitates the spatial divergence removal and the temporal divergence reduction through adversarial learning.

They show good performance on several action recognition benchmarks.

**Strengths:**

+ Unsupervised video domain adaptation presented in this paper is an interesting topic.

+ In general, the paper is well-written, well-organized.

+ Extensive experiments show the effectiveness of the proposed model.

+ There are also some nice visualizations and comparisons presented in the paper.

**Weaknesses:**

Major:

- The proposed method seems more like a combination of modules/blocks from existing works, the reviewer is not very clear to the following aspects (i) the complexity analysis of the proposed model w.r.t. existing unsupervised domain adaptation methods in action recognition (ii) the major difference / contribution compared to existing works (iii) why bother to design such a complicated model? The novelty is below the acceptance bar for this venue although the reviewer does like the experiments, comparisons and visualizations presented in the paper.

- Is the proposed model trained in an end-to-end way? Fig. 3 shows that there are many different components and different losses / regularizations, it is still not very clear to reviewer even though the final loss presented in Eq. (14) is a sum of different loss components.

- How do you choose the hyper-parameters for e.g., T, $\eta$, and a series of $\lambda$s for different loss components. The supplementary material only provides the parameters used in the experiments on different evaluations of different datasets, but (i) how to choose them is not mentioned (ii) analysis of the parameters is not presented and discussed, e.g., different magnitudes of the $\lambda$ for different sets of experiments presented in Table A of supplementary material (why for $\lambda_4$ the values are generally large).

- The choose of backbone is limited to I3D, which is published in CVPR'17, ~6 years ago, why not choosing the newest models for analysis? Many popular pipelines such as masked modeling (MAE for videos) and transformer-based video models are much more popular, are there any specific reasons of choosing I3D as backbone?

- Table 4 is not referenced in the paper? Table 4 is not being thoroughly analyzed and discussed. The reasons why existing methods achieve better performance on some experiments are not discussed in the paper. The reasons why other methods outperform the proposed model should be analyzed and discussed in the paper, e.g., with multi-modal signals, will the proposed model outperform the method CIA? Also it is suggested to conduct a complexity analysis among the proposed model versus the existing SOTA methods, e.g., the number of model parameters, processing speeds, FLOPs, etc.

Minor:

- It is suggested to have a notation section for the maths symbols used in the paper to make them clearer to readers. For example, regular fonts are scalars; vectors are denoted by lowercase boldface letters, matrices by the uppercase boldface, etc.

- Are there specific reasons to show both Fig. 5 and 6? They look almost the same and the only difference is: Fig. 5 for $\textbf{U} \rightarrow \textbf{H}$ and Fig. 6 is $\textbf{H} \rightarrow \textbf{U}$




**Questions:**

Please see my comments above.

I would consider raising scores if the authors are able to provide satisfactory responses to the raised concerns. .

**Limitations:**

Discussed in either the main submission or supplementary material.

---

> ### Author Rebuttal · Authors · 2023-08-09
>
> We thank Reviewer qCdJ for devoting time to this review and drawing valuable comments.
>
> ---
> > ***Q1:** "(i) Complexity analysis w.r.t. existing UDA methods in action recognition; (ii) the major difference/contribution compared to existing works; (iii) why bother to design such a complicated model?"*
>
> **A:** Thanks for your comment. We include a complexity analysis with existing video UDA methods on the number of trainable parameters, multiply-accumulate operations (MACs), floating-point operations (FLOPs), and inference frame-per-second (FPS) as follows.
> |Method|Trainable Params|MACs||FLOPs||FPS
> |-|:-:|-|-|-|-|-
> |CoMix|30.3688 M|18.5640 G||37.1280 G||0.0127 s
> |CO2A|23.6720 M|18.1884 G||36.3768 G||0.0157 s
> |TranSVAE|12.7419 M|18.2657 G||36.5314 G||0.0133 s
>
> We observe that TranSVAE requires less trainable parameters than prior arts. Since these approaches adopt the same I3D backbone, the MACs, FLOPs, and FPS are competitive among them.
>
> As for the differences to other models:
> - Firstly, we highlight that the primary idea of this paper is to provide a new perspective on solving video-based UDA problems, that is from a *generative view of cross-domain data*. To the best of our knowledge, this is the first study in video-based UDA. Using the disentanglement framework, Sequential VAE is just one way of achieving the generative process of cross-domain videos as shown in Figure 2.
> - Secondly, disentanglement methods like [26, 27, 28] aim to learn latent feature representation from input, where each component of learned features, usually including dynamic and static ones, refers to a semantically meaningful concept. However, disentanglement can be arbitrary, and the proposed TranSVAE is designed to obtain a specific disentanglement beneficial for the adaptation tasks. As shown in Figures 5 and 6, just using SVAE [26] for disentanglement, i.e. results (a), does not yield the optimal adaptation performance in video-based UDA.
> - Thirdly, in the TranSVAE framework, disentanglement and adaptation are interactive and complementary.  All the constraints serve to achieve a good disentanglement effect with the two-level domain divergence minimization. We would like to re-emphasize that the proposed contrastive triple loss ($L_{ctc}$) not only boosts the static consistency but also achieves good domain-specific latent factors by pulling close the inter-domain latent and pushing away the intra-domain ones, which contribute to the spatial domain alignment. We agree with the reviewer that some techniques, e.g. the $L_{mi}$ and $L_{adv}$ losses, are standard in their fields. However, the overall TranSVAE framework is not trivial as each component is not isolated but complementary for the adaptation purpose. The best adaptation performance is achieved with the interaction of all the loss terms as verified. The reviewer may refer to Table 3, Figures 5 and 6 in the paper for the ablation study of each loss term. More ablation analyses are also included in our response to other reviewers. Please kindly refer to the answer to Reviewer 3nXx's Q1, thanks.
>
> ---
> > ***Q2:** "Is the proposed model trained in an end-to-end way?"*
>
> **A:** Yes, TranSVAE is trained in an end-to-end way. The training process is quite standard as the conventional sequential VAE training. The difference is that we also need to a backward w.r.t. the losses specifically designed in this framework.
>
> ---
> > ***Q3:** "How do you choose the hyper-parameters for $T$, $\eta$, and a series of $\lambda$s for different loss components?"*
>
> **A:** Following existing video-based UDA methods [11,12,13,14,15,16,17], we use the grid search on the validation set to choose the hyper-parameters. An example of $T$ can be found in Table 2 of the Appendix. The essence of hyper-parameters is to balance the magnitudes of each loss term in the overall objective, and it is highly dependent on the training data. For instance, $\lambda_4$ is for the cross-entropy loss which is usually in the range of 1e-4 to 1e-1, and thus it is usually assigned with a large value to highlight the importance of the cross-entropy loss.
>
> ---
> > ***Q4:** "Are there any specific reasons for choosing I3D as backbone? Why not choose the newest models for analysis?"*
>
> **A:** We use I3D as the backbone following existing methods [11,12,13,14,15,16,17]; it is proven that I3D is powerful and efficient in handling different video-based UDA tasks. We agree with the reviewer that using the newest model as a backbone is a promising direction worthy of further study.
>
> ---
> > ***Q5:** "Table 4 is not being thoroughly analyzed and discussed. Also, it is suggested to conduct a complexity analysis among the proposed model versus the existing SOTA methods."*
>
> **A:** We are sorry for the non-reference of Table 4. It should be referenced in *"Section 4.3 - Compared to Mulit-Modal Methods"*.
> - Basically, it is not strictly fair to directly compare TranSVAE with multi-modality methods as we only use the single RGB as the input. We include the comparisons of Table 4 to show the promising results of TranSVAE even compared with those methods using more information, instead of aiming to beat them.
> - Currently, it is non-trivial to include multi-modality data in TranSVAE, and we take this as a promising future direction. However, from the current results, i.e., TranSVAE achieves comparable results with CIA, we believe it is promising to further boost the transfer performance if multi-modality data is considered in TranSVAE.
> - For more comparisons, kindly refer to Reviewer AobY's Q3, thanks.
>
> ---
> > ***Q6:** "It is suggested to have a notation section for the maths symbols used in the paper."*
>
> **A:** Thanks a lot for your suggestion. We have added such a table in our revised paper for better clarification of all the notations used.
>
> ---
> > ***Q7:** "Are there specific reasons to show both Fig. 5 and 6?"*
>
> **A:** Thanks for asking. This is to show the results of the ablation studies on both U -> H and H -> U tasks.

---

> > ### Comment · Reviewer_qCdJ · 2023-08-12
> >
> > Thanks for the detailed information.
> >
> > I want to know the following details before I adjust my rating towards final decision.
> >
> > (i) Q1 is nicely addressed, but for the use of old-fashioned I3D as backbone, do existing works that you listed all use I3D backbone only? Or are there other backbone being used?
> >
> > (ii) what kind of notions are going to be added, could you please show me?
> >
> > (iii) given Table 4 is not referenced, but now is going to be referenced in 'Section 4.3 - Compared to Multi-Modal Methods', are there extra discussions/comparisons that are interesting and worth to be added as well?
> >
> > (iv) what are the updates going to be made based on the above rebuttal in addressing the raised concerns (w.r.t. extra discussions, experiments and analysis etc)
> >
> > (v) with both Fig. 5 & 6, are there any extra information presented in both figures worth discussing or analysis apart from Line 323-330, or any interesting insights or comparisons to show readers (given two figures)?

---

> > > ### Author Response · Authors · 2023-08-13
> > > **Authors' Response to Reviewer qCdJ**
> > >
> > > We thank Reviewer qCdJ for devoting time to the Author-Reviewer discussion session and providing follow-up comments. Our responses are as follows.
> > >
> > > ---
> > > > ***Q-i:** "Do existing works that you listed all use I3D backbone only? Or are there other backbone being used?"*
> > >
> > > **A:** Thanks for the question.
> > > - Existing works that we are comparing with, including CO2A [13],  CoMix [14], MM-SADA [18], STCDA [19], CMCD [20], and CIA [21], all adopt I3D as their backbones.
> > > - Early works [11,12,16] used ResNet-101 as the backbone.
> > > - It has been widely verified in empirical ways that using the I3D backbone can achieve satisfactory performance across existing video-based UDA benchmarks, including those large-scale ones like UCF-HMDB and Epic-Kitchen.
> > > - We believe more powerful backbones are promising; as larger and higher-quality datasets come out, there will require an upgrade on the backbones for video-based UDA tasks.
> > >
> > > ---
> > > > ***Q-ii:** "What notions are going to be added, could you please show me?"*
> > >
> > > **A:** We omit the notation table in the previous rebuttal window due to the lack of space (there is a 6000-character limit). Specifically, we supplemented the following table in the revised manuscript to improve the readability:
> > > | Notation | Description
> > > |-|-
> > > |$\mathcal{D}$|Domain
> > > |$\mathcal{S}$ / $\mathcal{T}$|Source domain / Target domain
> > > |$\mathbf{V}^\mathcal{D}$|A video sequence from domain $\mathcal{D}$
> > > |$\{\mathbf{V}_i^\mathcal{D},y_i^\mathcal{D} \}$|The $i$-th video sequence and the corresponding action label of domain $\mathcal{D}$
> > > |$\{\mathbf{x}_i^\mathcal{D},…,\mathbf{x}_T^\mathcal{D} \}$|$T$ frames of images in the video sequence $\mathbf{V}^\mathcal{D}$
> > > |$\{\mathbf{z}_i^\mathcal{D},…,\mathbf{z}_T^\mathcal{D} \}$|The dynamic latent factors of the video sequence from domain $\mathcal{D}$
> > > |$\mathbf{z}_d^\mathcal{D}$|The static latent factors of the video sequence from domain $\mathcal{D}$
> > > |$\mathbf{x}_{<t}^\mathcal{D}$|The input sequence before timestamp $t$
> > > |$\mathbf{x}_{1:T}^\mathcal{D}$|The full input sequence from $t=1$ to $t= T$
> > >
> > > Please let us know if there is any other notation that is unclear and could be clarified. Thanks a lot.
> > >
> > > ---
> > > > ***Q-iii:** "'Are there extra discussions/comparisons in Sec. 4.3 that are interesting and worth to be added?"*
> > >
> > > **A:** Thanks for your question.
> > > - Section 4.3 *'Compared to Multi-Modal Methods'* is to compare and analyze TranSVAE with existing multi-modal UDA methods, and Table 4 is to show the corresponding comparison results. In the submitted version, we accidentally forgot to refer the table to the Section, while the analyses are already there.
> > > - In this revision, we have added comparisons and analyses on more multi-modal methods, including A3R, CydDA, MixDANN, and CleanAdapt, as suggested by *Reviewer cyvj* and *Reviewer AobY*. Please kindly check our responses to these reviewers for more details, thanks.
> > >
> > > ---
> > > > ***Q-iv:** "What are the updates going to be made based on the above rebuttal in addressing the raised concerns?"*
> > >
> > > **A:** Thanks for your question. We summarized the changes and improvements that we made during this rebuttal in the General Response section (copy below). We appreciate your valuable comments and suggestions. Please let us know if there is anything that we could further modify or improve. Thanks.
> > >
> > > - We have supplemented more ablation experiments, including TranSVAE w/ and w/o disentanglement, to verify the effectiveness of each objective term in our framework.
> > > - We have compared with more SOTA baselines that use multi-modality data sources for video-based UDA, including A3R, CleanAdapt, CycDA, and MixDANN. All the comparison results show that our TransVAE is still a competitive method even using only a single RGB modality.
> > > - We have added complexity analyses on the number of trainable parameters, multiply-accumulate operations, floating-point operations, and inference frame-per-second for our TranSVAE as well as for prior arts.
> > > - For better readability, we have supplemented a notation section for the maths symbols used in the paper.
> > > - We have added missing references suggested by the reviewers.
> > > - We have polished and improved the elaboration of this work.
> > > - We have carefully addressed other comments from the reviewers point by point.
> > >
> > > ---
> > > > ***Q-v:** "Any extra information in Fig. 5 & 6 worth discussing or analysis, or any interesting insights or comparisons to show readers?"*
> > >
> > > **A:** Thanks for your question.
> > > - The two figures which show the ablation study results on U/H datasets are indicating the effectiveness of each loss term in the TranSVAE framework. The reason why we showed two figures is for the completeness of U/H datasets, as some reviewers might be interested in the ablation results on both tasks.
> > > - As more contents and improvements have been supplemented in the revision, we have moved Fig. 6 to the Appendix due to limited space.
> > >
> > > ---
> > > Last but not least, we thank Reviewer qCdJ again for the time and effort devoted to this review.

---

> > > > ### Author Response · Authors · 2023-08-13
> > > > **References for the above Response**
> > > >
> > > > **References:**
> > > > - [13] V. G Turrisi, et al. “Dual-head contrastive domain adaptation for video action recognition.” *WACV*, 2022.
> > > > - [14] A. Sahoo, et al. “Contrast and mix: Temporal contrastive video domain adaptation with background mixing.” *NeurIPS*, 2021.
> > > > - [18] J. Munro and D. Damen. “Multi-modal domain adaptation for fine-grained action recognition.” *CVPR*, 2020.
> > > > - [19] X. Song, et al. “Spatiotemporal contrastive domain adaptation for action recognition.” *CVPR*, 2021.
> > > > - [20] D. Kim, et al. “Learning cross-modal contrastive features for video domain adaptation. *ICCV*, 2021.
> > > > - [21] L. Yang, et al. “Interact before align: Leveraging cross-modal knowledge for domain adaptive action recognition.” *CVPR*, 2022.
> > > > - [11] M.-H. Chen, et al. “Temporal attentive alignment for large-scale video domain adaptation.” *ICCV*, 2019.
> > > > - [12] Y. Luo, et al. “Adversarial bipartite graph learning for video domain adaptation.” *ACM MM*, 2020.
> > > > - [16] B. Pan, et al. “Adversarial cross-domain action recognition with co-attention.” *AAAI*, 2020.

---

> > > > > ### Comment · Reviewer_qCdJ · 2023-08-13
> > > > >
> > > > > Given the authors have provided enough details in addressing my raised concerns/questions/issues, I think now it is much clearer to me, hence I am happy to increase the rating.
> > > > >
> > > > > The authors should revise the paper carefully for a better version to match the high standards of NeurIPS quality.

---

> > > > > > ### Author Response · Authors · 2023-08-15
> > > > > > **Authors' Response to Reviewer qCdJ**
> > > > > >
> > > > > > We thank Reviewer qCdJ for acknowledging that our rebuttal is helping. As suggested, we will revise the manuscript with care.
> > > > > >
> > > > > > ---
> > > > > > Last but not least, we sincerely thank Reviewer qCdJ again for the time and effort devoted and the valuable comments drawn during this review.

---

### Official Review · Reviewer_AobY · 2023-07-05

**Soundness:** 2 fair
**Presentation:** 2 fair
**Contribution:** 2 fair
**Rating:** 5
**Confidence:** 5

**Summary:**

This paper proposes an unsupervised domain adaptive approach for video de-entangling based on unraveling spatial and temporal domain discrepancies. The proposed TranSVAE framework models the problem and optimizes the latent factors through multiple objective constraints. Experimental results demonstrate the effectiveness and superiority of this method in behavior recognition tasks.

**Strengths:**

1. This paper focuses on the unsupervised domain adaptation of videos, which is more challenging compared to images, as it considers not only spatial structures but also temporal relationships.

2. The motivation and writing of this paper are well done.

3. The experimental results show improvement compared to the baseline method.


**Weaknesses:**

1. It is mentioned that the approach of disentanglement for video tasks is relatively mature. Although the authors applied it to handle the domain adaptation problem by separating spatial and temporal domain differences, this approach may not be considered as novel.
2. It is noted that there is a curiosity about why the authors did not use the same backbone as CoMix for comparison experiments. It is suggested that using the same backbone for comparison would provide a fairer evaluation.
3. The baselines compared by the authors are considered outdated, as they include methods that are even two years old. It is recommended to include a comparison with more advanced state-of-the-art methods as following and analyze the differences.

[a] Yin, Yuehao, et al. "Mix-DANN and Dynamic-Modal-Distillation for Video Domain Adaptation." Proceedings of the 30th ACM International Conference on Multimedia. 2022.

[b] Lin, Wei, et al. "CycDA: Unsupervised Cycle Domain Adaptation to Learn from Image to Video." European Conference on Computer Vision. Cham: Springer Nature Switzerland, 2022.

4. Some minor:
(1) The clarity of Figure 3 is mentioned as not being sufficient to intuitively understand the overview of the authors' method.

(2) It would be beneficial to provide information about the inference time or FLOPs (floating-point operations) of the model for a more comprehensive understanding.


**Questions:**

1 Although the novelty of the proposed method in this paper is average, the experiments and ablation studies are comprehensive, and the work is solid.

2 I am particularly interested in the author's response to the second and third points raised in the weaknesses. If I am satisfied with the explanations, I may consider raising  score.


**Limitations:**

No needed.

---

> ### Author Rebuttal · Authors · 2023-08-09
>
> We thank Reviewer AobY for devoting time to this review and drawing valuable comments.
>
> ---
> > ***Q1:** "The approach of disentanglement for video tasks is relatively mature."*
>
> **A:** We would like to thank the reviewer’s comment.
> - Firstly, we highlight that the primary idea of this paper is to provide a new perspective on solving video-based UDA problems, that is from a generative view of cross-domain data. To the best of our knowledge, this is the first study in video-based UDA. Using the disentanglement framework, Sequential VAE is just one way of achieving the generative process of cross-domain videos as shown in Figure 2.
> - Secondly, disentanglement methods like [26, 27, 28] aim to learn latent feature representation from input, where each component of learned features, usually including dynamic and static ones, refers to a semantically meaningful concept. However, disentanglement can be arbitrary, and the proposed TranSVAE is designed to obtain a specific disentanglement beneficial for adaptation tasks. As shown in Figures 5 and 6, just using SVAE [26] for disentanglement, results (a), does not yield the optimal adaptation performance.
> - Thirdly, in the TranSVAE framework, disentanglement and adaptation are interactive and complementary. All the constraints serve to achieve good disentanglement with the two-level domain divergence minimization. We re-emphasize that the proposed CTC loss not only boosts the static consistency but also achieves good domain-specific latent factors by pulling close the inter-domain latent and pushing away the intra-domain ones, which contribute to the spatial domain alignment. We agree that some techniques, e.g. MI and ADV losses, are standard in their fields. However, the overall TranSVAE framework is not trivial as each component is not isolated but complementary for the adaptation purpose. The best adaptation performance is achieved with the interaction of all the loss terms. The reviewer may refer to Table 3, Figures 5 and 6 in the paper for the ablation study of each loss term. More ablation analyses are also included in our response to the Reviewer 3nXx.
>
> ---
> >  ***Q2:** "Why did not use the same backbone as CoMix for comparison experiments? It is suggested that using the same backbone for comparison would provide a fairer evaluation."*
>
> **A:** Thanks for the comment. We are actually using the exact same backbone I3D as CoMix. However, CoMix jointly trains the I3D network by first using the published pre-trained I3D parameters as a warm start and fine-tuning the whole weights; while our TranSVAE directly uses the generated features from published pre-trained I3D model without fine-tuning it.
>
> ---
> > ***Q3:** "It is recommended to include a comparison with more advanced state-of-the-art methods (Mix-DANN and CycDA) and analyze the differences."*
>
> **A:** Thanks for providing the reference works. We highlight the difference between Mix-DANN [R1] and CycDA [R2] from TranSVAE as follows. Both works utilize extra data sources except for RGB; flow in MixDANN and web data in CycDA, whereas TranSVAE only uses a single RGB modality. We have included these methods in the revision.
> |Method|D1->D2|D1->D3|D2->D1|D2->D3|D3->D1|D3->D2|Avg
> |-|-|-|-|-|-|-|-
> |MixDANN w/ flow|56.0|47.3|50.3|52.4|51.0|54.7|52.0
> |TranSVAE|50.5|50.3|50.3|58.6|48.0|58.0|52.6
>
> |Method|U->H|H->U|Avg
> |-|-|-|-
> |MixDANN w/o flow|77.5|86.5|82.0
> |MixDANN w/ flow|82.2|92.8|87.5
> |CycDA w/o web data|83.3|80.4|81.9
> |CycDA w/ web data|88.1|98.0|93.1
> |TranSVAE|87.78|98.95|93.37
>
> From the results of the U-H tasks, we can see that TranSVAE is better than MixDANN and CycDA if only a single RGB modality is used. TranSVAE is also better than the two methods even when multi-modality is used.
>
> For Epic-Kitchens, TranSVAE yields slightly better results than MixDANN with flow. Considering we only use a single modality, it shows the effectiveness of our framework for video UDA. Moreover, we believe introducing multi-modality data into the current framework is promising and worthy of further study. We will try to consolidate this in the revised paper.
>
> ---
> >***Q4:** "The clarity of Figure 3 is not sufficient to intuitively understand the overview of the method."*
>
> **A:** Thanks for the comment. We have included the following description of the TranSVAE framework in the revision.
> - The input videos are fed into an encoder to extract the visual features, followed by an LSTM to explore the temporal information.
> - Two groups of *mean* and *variance* networks are then applied to model the posterior of the latent factors, i.e., $q(z_t^{D}|x_{<t}^{D})$ and $q(z_d^{D}|x_{1:T}^{D})$.
> - The new representations $z_1^D,...,z_T^D$ and $z_d^{D}$ are sampled, and then concatenated and passed to a decoder for reconstruction. Four constraints are proposed to regulate the latent factors for adaptation.
>
> ---
> >***Q5:** "It would be beneficial to provide information about the inference time or FLOPs of the model for a more comprehensive understanding."*
>
> **A:** Thanks for the comment. We compare the model complexity with prior arts as follows:
> |Method|Trainable Params|MACs||FLOPs||FPS
> |-|:-:|-|-|-|-|-
> |CoMix|30.3688 M|18.5640 G||37.1280 G||0.0127 s
> |CO2A|23.6720 M|18.1884 G||36.3768 G||0.0157 s
> |TranSVAE|12.7419 M|18.2657 G||36.5314 G||0.0133 s
>
> We observe that TranSVAE requires less trainable parameters than prior arts. Since these approaches adopt the same I3D backbone, the MACs, FLOPs, and FPS are competitive among them.
>
> ---
> **References:**
> - [26] Disentangled sequential autoencoder. ICML 2018.
> - [27] S3VAE: Self-supervised sequential VAE for representation disentanglement and data generation. CVPR 2020.
> - [28] Contrastively disentangled sequential variational autoencoder. NeurIPS 2021.
> - [31] Domain-adversarial training of neural networks. JMLR 2016.
> - [R1] Mix-DANN and dynamic-modal-distillation for video domain adaptation. MM 2022.
> - [R2] CycDA: Unsupervised cycle domain adaptation to learn from image to video. ECCV 2022.

---

> > ### Comment · Reviewer_AobY · 2023-08-18
> >
> > After reading the reply from the authors, most concerns are solved, but it should add the revised baseline in revised version, so I choose to raise the score.

---

> > > ### Author Response · Authors · 2023-08-18
> > > **Authors' Response to Reviewer AobY**
> > >
> > > We thank Reviewer AobY for the positive feedback. As suggested, we will revise the manuscript accordingly based on the reviewers' comments and suggestions and incorporate every detail.
> > >
> > > ---
> > > Last but not least, we sincerely thank Reviewer AobY again for the time and effort devoted and the valuable comments drawn during this review.

---

### Official Review · Reviewer_cyvj · 2023-07-06

**Soundness:** 3 good
**Presentation:** 3 good
**Contribution:** 3 good
**Rating:** 5
**Confidence:** 4

**Summary:**

The paper proposes a Transfer Sequential VAE (TranSVAE) to model temporal and spatial domain divergence through feature disentanglement for unsupervised domain adaptation for action recognition from videos. The architecture consists of 5 components - encoder, LSTM, latent spaces, sampling and decoder. Video is generated using a VAE based structure from two sets of latent factors: Dynamic Random Variable to encode the semantic information for downstream tasks and Static Random Variable for domain-related spatial information. To make disentanglement facilitate adaptation, the latent factors are constrained following specific criteria. The approach is tested on UCF-HMDB, Jester and Epic Kitchens dataset and outperforms the concurrent related work by a significant margin.

**Strengths:**

●	The approach of using static and dynamic random variable to encode semantic and domain-related spatial information using the TranSVAE architecture is interesting

●	Results are comprehensive, and well elaborated on multiple datasets, consistent performance improvements over previous methods of single and multi-modal data.

●	Comprehensive ablation study shows the utility of the loss functions deployed.

**Weaknesses:**

●	Summarizing the main contributions point by point would make the paper a better read

●	[Section 3] A more detailed description of the architecture and input, output of each component would have made it easier to understand the architecture

●	Methodology needs to rewritten for better explainability. (See Questions: 1 LINE 115)

●	[Section 4.3] Providing quantitative results with ResNet-101 as backbone would have proved the effectiveness of using I3D over ResNet-101 on the TranSVAE architecture

●	Ablation experiment without disentanglement would justify the necessity of disentanglement.

●	Recent results are missing A3R [1] for Epic-Kitchens. CleanAdapt [2], CycDA [3] for U-> H H->U.

[1]: Audio-adaptive activity recognition across video domains. CVPR’22.
[2]: Overcoming Label Noise for Source-free Unsupervised Video Domain Adaptation. ICVGIP’22,
[3]: CycDA: Unsupervised Cycle Domain Adaptation to Learn from Image to Video, ECCV’22

Minor Weakness: static variable z_d & dynamic z_t should be written in Video Sequence Reconstruction Line 124-125.
Equation 1,2,3 can be explained much better if the formulas were derived.

**Post rebuttal**
Thank you authors, the response satisfactory answers the questions I had.
The rebuttal shows more results (lot of comparisons missing in original submission, Recent SOTA comparison, and FLOPs comparison). Hopefully these will be added to revised paper as well. Better expiation of methodology in order to understand the pipeline (more annotations in the figure, table explaining symbols, etc.), has been provided.

**Questions:**

●	Line 115 Framework Overview is not clear. Z_1 , ..., z_T and z_d. What do they mean. Whats ”d” here? What does x_{<t} and x_{1:T} mean at LINE 118. Line 123 again uses it without explaining what do they mean/represent/ are obtained?

●	LSTM module is unclear. From Figure 3, LSTM (yellow) seems to indicate all the frames (from both videos) are aware of one another and module looks like Fully connected network, while from description 127-129 It seems like the same LSTM applies separately to the source and target videos.

●	[Section 3] Since there is no mention of reconstruction loss, how is the decoder getting trained?

●	In most of the cases (Tab 1,2) the accuracy of supervised targets (T_sup) is higher than the TranSVAE, so does that mean the target pseudo-labels used in TranSVAE for all the datasets are almost correct?

●	Why some of the recent results missing? TRN [32],  CIA [21]. for H->U, and U->H. CIA [21] for epic kitchen.

**Limitations:**

Yes, provided.

---

> ### Author Rebuttal · Authors · 2023-08-09
>
> We thank Reviewer cyvj for devoting time to this review and drawing valuable comments.
>
> ---
> > ***Q1:** "Summarizing the main contributions point by point would make the paper a better read."*
>
> **A:** We have included the following summarization in the revision. The main contribution can be summarized as follows:
> - We provide a new perspective on solving video-based UDA problems, that is from a generative view of cross-domain videos. We develop a generative graphical model for the cross-domain video generation process and propose to utilize the sequential VAE as the base generative model.
> - Based on the above generative view, we propose a TranSVAE framework for video-based UDA. By developing four constraints on the latent factors to enable disentanglement to benefit adaptation, the proposed framework is capable of handling the cross-domain divergence from both spatial and temporal levels.
> - We conduct extensive experiments on several benchmark datasets to verify the effectiveness of TranSVAE. Comprehensive ablation study also demonstrates the positive effect of each loss term on adaptation.
>
> ---
> > ***Q2:** "A more detailed description of the input and output of each component would have made it easier to understand the architecture."*
>
> **A:**  We have included the following description in the revision.
> - The input videos are fed into an encoder to extract the visual features, followed by an LSTM to explore the temporal information.
> - Two groups of *mean* and *variance* networks are then applied to model the posterior of the latent factors, i.e., $q(z_t^{D}|x_{<t}^{D})$ and $q(z_d^{D}|x_{1:T}^{D})$.
> - The new representations $z_1^D,...,z_T^D$ and $z_d^{D}$ are sampled, and then concatenated and passed to a decoder for reconstruction. Four constraints are proposed to regulate the latent factors for adaptation.
>
> ---
> > ***Q3:** "Better explainability of lines 115, 118, and 123."*
>
> **A:** We give the meaning of basic notations in the Introduction, lines 46-49. As suggested, we have added a table for better clarification of all the notations. $z_1, ..., z_T$ are the dynamic latent representations. $z_d$ is the static latent representation, where "$d$" is the abbreviation of "domain" as it is expected to be domain-specific. $x_{<t}$ represents the input sequence before time $t$, and $x_{1:T}$ represents the full sequence from $t=1$ to $t=T$.
>
> ---
> > ***Q4:** "Results with ResNet-101 as backbone."*
>
> **A:** We have included the TransVAE results of ResNet-101 in the revision.
> |Backbone|U->H|H->U|Avg|
> |-|-|-|-|
> |Res101|81.94|88.44|85.19
> |I3D|87.78|98.95|93.37
>
> ---
> > ***Q5:** "Ablation experiment without disentanglement would justify the necessity of disentanglement."*
>
> **A:** Thanks for the comment. We present the analysis of each loss term w/o disentanglement, i.e., not adding $L_{svae}$, as follows:
> |-|$L_{svae}$|$L_{adv}$|$L_{mi}$|$L_{ctc}$|Acc (U->H)|Acc (H->U)
> |-|-|-|-|-|:-:|:-:|
> |$\mathcal{S}_{only}$|||||80.27|88.79
> |-||✓|||81.67|90.54
> |-|||✓||79.44|90.37
> |-||||✓|81.11|90.37
> |-||✓|✓|✓|82.22|91.07
> |-|✓|✓|✓|✓| 87.78|98.95
>
> Without $L_{svae}$, the framework is to learn representations with corresponding constraints as many existing video-based UDA methods do. Due to the space limit, please kindly refer to Reviewer 3nXx's Q1 for more detailed explanations of the different disentanglement effects.
>
> ---
> > ***Q6:** "Recent results of A3R for Epic-Kitchens, CleanAdapt and CycDA for U-H."*
>
> **A:** Thanks for providing the reference works. We highlight the difference between these works from TranSVAE. All three works utilize *extra data sources* except for RGB, specifically audio in A3R, flow in CleanAdapt, and web data in CycDA, where current TranSVAE only uses a single RGB modality. We have included these methods for a more comprehensive comparison in the revision.
> |Method|D1->D2|D1->D3|D2->D1|D2->D3|D3->D1|D3->D2|Avg
> |-|-|-|-|-|-|-|-
> |A3R|53.2|52.1|51.9|55.5|48.7|63.2|54.1
> |CleanAdapt (2-stream)|52.7|47.0|46.2|52.7|47.8|54.4|50.3
> |TranSVAE|50.5|50.3|50.3|58.6|48.0|58.0|52.6
>
> |Method|U->H|H->U|Avg
> |-|-|-|-
> |CleanAdapt|86.1|96.1|91.1
> |CleanAdapt (2-stream)|89.8|99.2|94.5
> |CycDA w/o web data|83.3|80.4|81.9
> |CycDA w/ web data|88.1|98.0|93.1
> |TranSVAE|87.78|98.95|93.37
>
> From the results of U/H tasks, we can see that TranSVAE is better than CleanAdapt and CycDA if only a single RGB modality is used in these corresponding methods. TranSVAE is even better than CycDA and is around 1.1% worse than CleanAdapt if multi-modality is used.
>
> For Epic-Kitchens, TranSVAE is better than CleanAdapt and is 1.5% worse than A3R where both baselines use multi-modality data. Considering TranSVAE only uses a single modality, it shows the effectiveness of TranSVAE for video-based UDA. Moreover, we believe introducing multi-modality data into TranSVAE is a promising direction and will try it during the revision.
>
> ---
> > ***Q7:** "$z_d$ & $z_t$ should be written in line 124-125. Derived Eq. 1,2,3."*
>
> **A:** Thanks for the suggestion. We have revised them accordingly.
>
> ---
> > ***Q8:** "LSTM module in Fig.3 is unclear."*
>
> **A:** The LSTM module is shared by two domains. Fig. 3 is only for visualization and we show the detailed structure in Section 1.2 of the Appendix.
>
> ---
> > ***Q9:** "How is the decoder getting trained?"*
>
> **A:** The reconstruction loss is already included in $L_{svae}$, i.e. Eq. 4.
>
> ---
> > ***Q10:** "$T_{sup}$ is higher than TranSVAE, does that mean the target pseudo-labels used in TranSVAE for all the datasets are almost correct?"*
>
> **A:** $T_{sup}$ means directly using target labeled data for training, and thus can be taken as the upper bound of the adaptation performance. This is a conventional baseline widely used in existing video-based UDA methods, e.g. [11-16].
>
> ---
> > ***Q11:** Results of TRN and CIA are missing."*
>
> **A:** Thanks for the comment. TRN is not a video UDA method and thus is not compared. CIA is a video UDA method using multi-modality data, we already included the results of CIA in Tab. 4.

---

> > ### Comment · Reviewer_cyvj · 2023-08-18
> >
> > Thank you authors, the response satisfactory answers the questions I had.
> > The rebuttal shows more results (lot of comparisons missing in original submission, Recent SOTA comparison, and FLOPs comparison). Hopefully these will be added to revised paper as well. Better expiation of methodology in order to understand the pipeline (more annotations in the figure, table explaining symbols, etc.), has been provided.
> > I have no further questions for the authors.

---

> > > ### Author Response · Authors · 2023-08-18
> > > **Authors' Response to Reviewer cyvj**
> > >
> > > We thank Reviewer cyvj for acknowledging that our rebuttal is helpful. As suggested, we will revise the manuscript accordingly based on the reviewers' comments and suggestions.
> > >
> > > ---
> > > Last but not least, we sincerely thank Reviewer cyvj again for the time and effort devoted and the valuable comments drawn during this review.

---

### Official Review · Reviewer_3nXx · 2023-07-06

**Soundness:** 4 excellent
**Presentation:** 3 good
**Contribution:** 3 good
**Rating:** 5
**Confidence:** 4

**Summary:**

This paper focuses on video domain adaptation. The authors' primary focus is to eliminate the temporal invariant component of the features initially, followed by the implementation of an adversarial-based domain adaptation technique. To achieve disentanglement, they employ a method of temporally random shuffling of the video, thus identifying domain-specific and static latent factors that remain unaffected by temporal shifts. These domain static factors, along with the Domain-Invariant Sequential Factors, are subsequently utilized separately for domain adaptation. The authors provide an extensive evaluation of their proposed TranSVAE method by conducting rigorous experiments on widely recognized datasets such as UCF-HMDB, Jester, and Epic-Kitchens. The results demonstrate that the TranSVAE model surpasses several state-of-the-art techniques, including those employing multi-modal approaches.




**Strengths:**

1. The proposed idea of handling spatial and temporal domain divergence separately through disentanglement for domain adaptation is interesting.
2. The idea of using shuffling to disentangle the domain static and dynamic latent factors is interesting.
2. The method can outperform sota with only RGB modality on multiple datasets.

**Weaknesses:**

1. The proposed model contains many components but no proper ablation study is conducted to evaluate the effect of each component. In my opinion, the core part should be whether to add $L_{mi}$ and/or $L_{ctc}$ into the model. The ablation study in Figures 5 and 6 is not sufficient. The authors should add ablation study to specifically evaluate this core part, without the influence of other components of the model, especially $L_{svae}$.
2. While this method outperforms previous works, it apparently uses much more parameters because of the use of a decoder. An analysis of training and inference time and model parameters is needed.
3. The EPIC-Kithens dataset has a new official split for domain adaptation, which is a larger and better split with an online evaluation server for the test set. This split is used in a lot of previous works. I wonder why the authors do not perform experiments on this set.

**Questions:**

1. I hope the authors can provide better ablation study for better highlighting the core technical part of this work. In my opinion, the use of $L_{svae}$, $L_{cls}$, $L_{adv}$, and PL are not the main technical contribution of this work.
2. I hope the authors can provide a comparative analysis on training time, inference time, and model parameters.
3. The EPIC-Kitchens100 dataset should be a better playground for evaluating this method.

**Limitations:**

Limitations are discussed by the authors in the paper.

---

> ### Author Rebuttal · Authors · 2023-08-09
>
> We thank Reviewer 3nXx for devoting time to this review and drawing valuable comments.
>
> ---
> > ***Q1:** "Add an ablation study to specifically evaluate $L_{mi}$ and/or $L_{ctc}$, without the influence of other components, especially $L_{svae}$."*
>
> **A:** Thanks for your suggestion. We highlight that the primary idea of this work is providing a generative perspective on solving video-based UDA problems. Using Sequential VAE is one way of achieving the generative process of cross-domain videos as shown in Figure 2, and thus $L_{svae}$ is indispensable in the TranSVAE framework. Moreover, in our framework, disentanglement and adaptation are interactive and complementary. All the losses work together to obtain a specific disentanglement beneficial for adaptation.
>
> We agree with the reviewer that more ablation analyses will further solidify our work, and thus we conduct the following experiments. Note that here $L_{cls}$ is always needed as it is used to train the final classification model.
>
> 1. Analyzing the effect of each loss term w/o disentanglement, i.e., not adding $L_{svae}$:
> |-|$L_{svae}$|$L_{adv}$|$L_{mi}$|$L_{ctc}$|Acc (U->H)|Acc (H->U)
> |-|-|-|-|-|:-:|:-:|
> |$\mathcal{S}_{only}$|||||80.27|88.79
> |-||✓|||81.67|90.54
> |-|||✓||79.44|90.37
> |-||||✓|81.11|90.37
> |-||✓|✓|✓|82.22|91.07
> |-|✓|✓|✓|✓| 87.78|98.95
> - Without $L_{svae}$, the framework is to learn representations with corresponding constraints as many existing video-based UDA methods do. Precisely, we have the following remarks:
>   - Only with $L_{adv}$, the framework is equivalent to TA3N [11], which learns the domain-invariant temporal representation. We can see the adaptation results are similar to TA3N.
>   - $L_{mi}$ is a term specifically designed for disentanglement purposes. Thus, the effect of using $L_{mi}$ alone for adaptation is random as verified in the above table, achieving worse results in the U -> H task and better results in the H -> U tasks compared with the source-only baseline.
>   - $L_{ctc}$ enforces the learned static representation to be domain-specific and brings indirect benefit to adaptation, thus obtaining slightly better results than the source-only baseline.
> - In sum, without $L_{svae}$, the adaptation performance of each loss term alone is far from optimal. Moreover, combining all the terms can slightly improve the adaptation performance over the individual ones, however, is still obviously worse than TranSVAE results. The above ablation study shows the necessity of the disentanglement in the TranSVAE framework and proves that $L_{svae}$ is an indispensable loss term.
>
> 2. Analyzing the effect of each loss term w/ disentanglement:
> |-|$L_{svae}$|$L_{adv}$|$L_{mi}$|$L_{ctc}$|Acc (U->H)|Acc (H->U)
> |-|-|-|-|-|:-:|:-:|
> |$\mathcal{S}_{only}$|||||80.27|88.79
> |-|✓|✓|||84.44|93.52
> |-|✓||✓||83.06|93.70
> |-|✓|||✓|83.61|91.42
> |-|✓|✓|✓|✓| 87.78|98.95
> - From these results, we have the following remarks:
>   - Each individual term w/ $L_{svae}$ achieves better adaptation results than the source-only baseline.
>   - Each individual term w/ $L_{svae}$ consistently outperforms the corresponding ones w/o $L_{svae}$.
>   - The final TranSVAE combined all the loss terms yields much better results than each individual case.
> - This set of ablation studies indicates that 1) each term in TranSVAE brings positive effects to adaptation; 2) disentanglement and adaptation are interactive and complementary in our framework to obtain the best possible UDA results.
>
> 3. Analyzing the effect of removing each loss term:
> |-|$L_{svae}$|$L_{adv}$|$L_{mi}$|$L_{ctc}$|Acc (U->H)|Acc (H->U)
> |-|-|-|-|-|:-:|:-:|
> |$\mathcal{S}_{only}$|||||80.27|88.79
> |-|✓||✓|✓|83.06|93.52
> |-|✓|✓||✓|85.83|93.52
> |-|✓|✓|✓||83.89|95.80
> |-|✓|✓|✓|✓| 87.78|98.95
> - From the above results, we can see that removing any of the terms leads to inferior results than the full TranSVAE, and removing $L_{adv}$ is the most influential. This is reasonable as $L_{adv}$ is used to explicitly reduce the temporal domain gaps. All these results show that each loss term matters in the proposed TranSVAE framework.
> - As suggested, we have included all these ablation results and analyses in the revision.
>
> ---
> > ***Q2:** "An analysis of training and inference time and model parameters is needed."*
>
> **A:** Thanks for the comment. We compare the number of trainable parameters, multiply-accumulate operations (MACs), floating-point operations (FLOPs), and inference frame-per-second (FPS) with prior arts as follows:
> |Method|Trainable Params|MACs||FLOPs||FPS
> |-|:-:|-|-|-|-|-
> |CoMix|30.3688 M|18.5640 G||37.1280 G||0.0157 s
> |CO2A|23.6720 M|18.1884 G||36.3768 G||0.0127 s
> |TranSVAE|12.7419 M|18.2657 G||36.5314 G||0.0133 s
>
> We observe that TranSVAE requires less trainable parameters than prior arts. Since these approaches adopt the same I3D backbone, the MACs, FLOPs, and FPS are competitive among them.
>
> ---
> > ***Q3:** "Perform experiments on the new official split of EPIC-Kithens domain adaptation."*
>
> **A:** We thank the reviewer for informing us about the new version of this dataset. In this work, we adopt the benchmark EPIC-Kithens dataset and splits as widely used in the existing video-based UDA methods, e.g. [2,11,14,18], as well as some most recent works [R1,R2]. As suggested, we will test this new split and report the results in the revised version.
>
> ---
> **References:**
> - [2] E. Tzeng, et al. Adversarial discriminative domain adaptation. CVPR, 2017.
> - [11] M.-H. Chen, et al. Temporal attentive alignment for large-scale video domain adaptation. ICCV, 2019.
> - [14] A. Sahoo, et al. Contrast and mix: Temporal contrastive video domain adaptation with background mixing. NeurIPS, 2021.
> - [18] J. Munro and D. Damen. Multi-modal domain adaptation for fine-grained action recognition. CVPR, 2020.
> - [R1] A Dasgupta, et al. Overcoming label noise for source-free unsupervised video domain adaptation. ICVGIP, 2022.
> - [R2] Y. Zhang, et al. Audio-adaptive activity recognition across video domains. CVPR, 2022.

---

> > ### Comment · Reviewer_3nXx · 2023-08-14
> > **follow up questions for the authors**
> >
> > Many thanks to the author's response. I have the following questions regarding the response.
> > 1. The calculation of parameters and flops. I believe the authors calculated the MACs, FLOPs, and FPS of the inference stage, thus the complexity of the decoder is not counted. What about the trainable parameters? Are the parameters of the decoder counted? Can you also list the statistics of [11] just for reference?
> > 2. From the provided table, it is clear that the major performance gain comes from $L_{svae}$. Since the reconstruction process is conceptually similar to methods like MAE and VideoMAE, as also mentioned by qCdJ I think it is important to switch the backbone to the MAE based models.

---

> > > ### Author Response · Authors · 2023-08-15
> > > **Authors' Response to Reviewer 3nXx**
> > >
> > > We thank Reviewer 3nXx for devoting time to the Author-Reviewer discussion session and providing follow-up comments. Our responses are as follows.
> > >
> > > ---
> > > > ***Q-i:** "For the calculation of parameters and flops: are the parameters of the decoder counted? Can you also list the statistics of [11] just for reference?"*
> > >
> > > **A:** Thanks for your question.
> > > - Yes, all trainable parameters are counted in the calculation, including the parameters of the TranSVAE decoder. The statistics of TA3N [11] with the I3D backbone are shown as follows:
> > > |Method|Trainable Params||MACs|FLOPs|FPS||U->H$\uparrow$|H->U$\uparrow$|Average$\uparrow$|
> > > |-|:-:|-|-|-|-|-|-|-|:-:
> > > |CoMix|30.3688 M||18.5640 G|37.1280 G|0.0157 s||86.66|93.87|90.22
> > > |CO2A|23.6720 M||18.1884 G|36.3768 G|0.0127 s||87.78|95.79|91.79
> > > |TA3N|7.6880 M||18.2318 G|36.4636 G|0.0134 s||81.38|90.54|85.96
> > > |TranSVAE|12.7419 M||18.2657 G|36.5314 G|0.0133 s||87.78|98.95|93.37
> > >
> > > ---
> > > > ***Q-ii:** "(a) From the provided table, it is clear that the major performance gain comes from $L_{svae}$. (b) Since the reconstruction process is conceptually similar to methods like MAE and VideoMAE, it is important to switch the backbone to the MAE-based models."*
> > >
> > > **A:** Thanks for your questions.
> > > - For question (a):
> > >   - $L_{svae}$ is an indispensable component in TranSVAE. However, it is the interaction and combination of all the optimization constraints that provide the main adaptation performance gain. As shown in Figures 5 and 6, just using $L_{svae}$ for disentanglement, i.e. results (a) U-> H 82.50 and H-> U 92.29, does not yield the optimal adaptation performance in the video-based UDA tasks.
> > > - For question (b):
> > >   - We are sorry for the confusion on the terminology *"backbone"*. We would need to clarify that the term *"backbone"* used in the paper is not referring to the base model structure of TranSVAE. The reviewer might want to refer to Section 1.2 of the Appendix for the detailed architectures of TranSVAE.
> > >   - In contrast, *"backbone"* here refers to the model used to *pre-process* the data, as widely followed by prior video-based UDA approaches [11,14,15,16]. In existing methods, the video sequences are first pre-processed by either ResNet-101 [11] or I3D [15], where this pre-processing step was specially designed based on these backbone models. It has been widely validated that these backbones can provide reliable baseline performance for video-based UDA tasks.
> > >   - Going back to TranSVAE and VideoMAE [R1,R2], although both TranSVAE and VideoMAE are encoder-decoder structures, their mechanisms differ considerably.
> > >   - TranSVAE models the *posterior distribution* of the *latent variables*. The latent embeddings are sampled from the posterior distribution, and then fed into the decoder. The learning objective of TranSVAE consists of two parts: *i)* reconstruction loss and *ii)* KL divergence loss, where the *latter* is to align the *posterior* and *prior* distributions of latent variables and plays an important role in enabling the disentanglement of latent variables.
> > >   - VideoMAE, on the other hand, maps input data (e.g., video frames and audio) into a shared embedding space and reconstructs the original data from this embedding space, enabling cross-modal reconstruction. However, it does not model the data distribution and there is no sampling process inside.
> > > - We would like to further clarify that:
> > >   - The primary idea of this paper is to provide a *generative perspective* on solving video-based UDA problems. Using Sequential VAE is one way of achieving the generative process of cross-domain, and it has been empirically verified that disentanglement and adaptation are interactive and complementary to obtain the optimal adaptation performance.
> > > It is not trivial to adapt VideoMAE [R1,R2] to the above idea as we need to specifically re-design its structure to enable disentanglement.
> > >   - However, it might be possible to apply VideoMAE [R1,R2] to *pre-process* videos and then use its output for the following UDA tasks. By doing so, VideoMAE can be another *"backbone"* for video-based UDA. We have made a preliminary try by using a similar pre-processing step of I3D to VideoMAE feature extraction. We found that TranSVAE with such a VideoMAE feature does not obtain satisfactory results on U-H datasets, as shown below for your reference.
> > > |Source-Only|VideoMAE|I3D
> > > |-|-|-
> > > |U -> H|60.83|80.27
> > > |H -> U|61.39|88.79
> > >   - We believe it would require a deliberate design of the pre-processing step using VideoMAE [R1,R2] as the backbone. We are confident that this can be a promising direction worthy of future study.
> > >
> > > ---
> > > Last but not least, we thank Reviewer 3nXx again for the time and effort devoted to this review.

---

> > > > ### Author Response · Authors · 2023-08-15
> > > > **References for the above Response**
> > > >
> > > > **References:**
> > > > - [11] Min-Hung Chen, et al. "Temporal attentive alignment for large-scale video domain adaptation." *ICCV*, 2019.
> > > > - [14] Aadarsh Sahoo, et al. "Contrast and mix: Temporal contrastive video domain adaptation with background mixing." *NeurIPS*, 2021.
> > > > - [15]  Jinwoo Choi, et al. "Shuffle and attend: Video domain adaptation." *ECCV*, 2020.
> > > > - [16] Boxiao Pan, et al. "Adversarial cross-domain action recognition with co-attention." *AAAI*, 2020.
> > > > - [R1] Zhan Tong, et al. "VideoMAE: Masked autoencoders are data-efficient learners for self-supervised video pre-training." *NeurIPS*, 2022.
> > > > - [R2] Limin Wang, et al. "VideoMAE V2: Scaling video masked autoencoders with dual masking." *CVPR*, 2023.

---

> > > > ### Comment · Reviewer_3nXx · 2023-08-18
> > > >
> > > > Thanks to the authors for their reply.
> > > > I still have some follow up questions/requests about these two points.
> > > > 1. I can see that the listed computational cost is almost the same across all methods. Is it because the inference stage are all similar across these models, and this computational cost is calculated only for the inference stage?
> > > >
> > > > 2. I understand the authors' terminology that "backbone" here refers to the model used to pre-process the data. I mean that the encoder of VideoMAE can be the backbone to pre-process the data (simply for feature extraction). As I see in the Table in the reply, the performance is surprisingly bad. While I don't believe this performance gap needs some specific design since the training also contains the classification loss (it may mainly come from the choice of hyperparameters), I think this table should be included in the main manuscript and at least discussed more thoroughly about the reason.

---

> > > > > ### Author Response · Authors · 2023-08-18
> > > > > **Authors' Response to Reviewer 3nXx**
> > > > >
> > > > > We thank Reviewer 3nXx for the follow-up questions.
> > > > >
> > > > > ---
> > > > > For question 1:
> > > > > - Yes, existing video-based UDA approaches share a similar inference stage. Since all listed methods adopted the same backbone, their computational costs (at the inference stage) are similar.
> > > > > - In fact, existing UDA focused more on improving the performance of the target domain, rather than efficiency. We agree with the reviewer that the computational overhead should also be an important indicator and will encourage follow-up works to pay more attention to this aspect.
> > > > >
> > > > > ---
> > > > > For question 2:
> > > > > - We agree with the reviewer that conducting such a preliminary study (shifting the old backbone to VideoMAEs [R1,R2]) is interesting and will include the results in the revised manuscript.
> > > > > - We believe this kind of study could open up new possibilities for future research on video-based UDA. We will try to set up the baselines on both VideoMAE [R1] and VideoMAE V2 [R2] and will encourage follow-up works to incorporate them as the backbone models for video-based UDA. Thanks again for your suggestion.
> > > > >
> > > > > ---
> > > > > **References:**
> > > > > - [R1] Zhan Tong, et al. "VideoMAE: Masked autoencoders are data-efficient learners for self-supervised video pre-training." *NeurIPS*, 2022.
> > > > > - [R2] Limin Wang, et al. "VideoMAE V2: Scaling video masked autoencoders with dual masking." *CVPR*, 2023.
> > > > >
> > > > > ---
> > > > > Last but not least, we thank Reviewer 3nXx again for the time and effort devoted to this review.

---

### Official Review · Reviewer_JyS2 · 2023-07-06

**Soundness:** 3 good
**Presentation:** 4 excellent
**Contribution:** 2 fair
**Rating:** 6
**Confidence:** 4

**Summary:**

The paper tackles the unsupervised domain adaptation (UDA) problem by viewing it from a disentanglement perspective. It suggests separating two sets of latent factors, one of which is responsible for domain-related spatial information and another encodes dynamic temporal information. As a result, a new model for UDA called Transfer Sequential VAE (TranSVAE) is introduced. To achieve better disentanglement and better learning the paper considers additional constraints: minimizing mutual dependence, task-specific supervision, static consistency, and temporal domain alignment through adversarial-based idea [31]. The experiment set includes 4 dataset settings: UCF-HMDB, Jester, Epic-Kitchens, and Sprites. The proposed method outperforms related works on all these settings visually showing strong disentanglement properties on the Sprites dataset.

**Strengths:**

1) The paper is well-written with a good structure and useful visualizations. The motivation of the method is clear and easy to understand and follow.
2) The method achieves good performance on several datasets including real-world domains (UCF-HMDB, Jester, Epic-Kithens) and synthetic domains (Sprites). Additionally, it is visually shown a good disentanglement property on the Sprites dataset and analysis of different loss parts to study the domain and semantic alignment of the features.
3) The ablation study extensively analysis the impact of different proposed constraints showing that incorporating all of the losses achieves the best results.
4) The proposed method is also performing on par with multi-modal methods that are using other modalities such as optical flow. Sometimes it even outperforms most of them.

**Weaknesses:**

I would like to see more discussions about related works in the disentanglement area and more outlined explanations of differences of the proposed TranSVAE from them, especially with the papers for video generation [26, 27, 28]. Basically, they also consider the separation of static and dynamic latent factors and considers the minimization of mutual information between them. So it seems that the main difference lies in UDA setting for video action recognition itself (task-specific supervision) and other constraints in consideration (static consistency and temporal domain alignment). The temporal domain alignment is based on the existing ideas of [31], so the paper looks to me mostly as a smart combination of the existing approaches not as a fully novel model.

**Questions:**

I have no additional questions except the Weakness section discussion.

**Limitations:**

The paper adequately addressed the limitations and potential negative social impact of their work.

---

> ### Author Rebuttal · Authors · 2023-08-08
>
> We thank Reviewer JyS2 for devoting time to this review and drawing valuable comments.
>
> ---
> > ***Q:** "More discussions about related works in the disentanglement area and more outlined explanations of differences of the proposed TranSVAE from them, especially with the papers for video generation [26, 27, 28], which also consider the separation of static and dynamic latent factors and consider the minimization of mutual information between them."*
>
> **A:** We would like to thank the reviewer’s comment. Our response to this question is as follows:
> - Firstly, we highlight that the primary idea of this paper is to provide a new perspective on solving video-based UDA problems, that is from a *generative view of cross-domain data*. To the best of our knowledge, this is the first study in video-based UDA. Using the disentanglement framework, Sequential VAE is just one way of achieving the generative process of cross-domain videos as shown in Figure 2.
> - Secondly, disentanglement methods like [26, 27, 28] aim to learn latent feature representation from input, where each component of learned features, usually including dynamic and static ones, refers to a semantically meaningful concept. However, disentanglement can be arbitrary, and the proposed TranSVAE is designed to obtain a specific disentanglement beneficial for the adaptation tasks. As shown in Figures 5 and 6, just using SVAE [26] for disentanglement, i.e. results (a), does not yield the optimal adaptation performance in video-based UDA.
> - Thirdly, in the TranSVAE framework, disentanglement and adaptation are interactive and complementary.  All the constraints serve to achieve a good disentanglement effect with the two-level domain divergence minimization. We would like to re-emphasize that the proposed contrastive triple loss ($L_{ctc}$) not only boosts the static consistency but also achieves good domain-specific latent factors by pulling close the inter-domain latent and pushing away the intra-domain ones, which contribute to the spatial domain alignment. We agree with the reviewer that some techniques, e.g. the $L_{mi}$ and $L_{adv}$ losses, are standard in their fields. However, the overall TranSVAE framework is not trivial as each component is not isolated but complementary for the adaptation purpose. The best adaptation performance is achieved with the interaction of all the loss terms as verified. The reviewer may refer to Table 3, Figures 5 and 6 in the paper for the ablation study of each loss term. More ablation analyses are also included in our response to other reviewers. Please kindly refer to the answer to Reviewer 3nXx's Q1, thanks.
>
> ---
> **References:**
> - [26] Y. Li and S. Mandt. Disentangled sequential autoencoder. ICML, 2018.
> - [27] Y. Zhu, et al. S3VAE: Self-supervised sequential VAE for representation disentanglement and data generation. CVPR, 2020.
> - [28] J. Bai, et al. Contrastively disentangled sequential variational autoencoder. NeurIPS, 2021.
> - [31] Y. Ganin, et al. Domain-adversarial training of neural networks. JMLR, 2016.

---

> > ### Comment · Reviewer_JyS2 · 2023-08-20
> > **Thank you for the Rebuttal**
> >
> > Thank you for the clear and detailed response to my question. I have a favorable opinion of the paper that seems aligned with other reviewers. I will keep my initial "Weak Accept" rating.

---

> > > ### Author Response · Authors · 2023-08-20
> > > **Authors' Response to Reviewer JyS2**
> > >
> > > We sincerely thank Reviewer JyS2 for the positive feedback provided and the time and effort devoted during this review.

---

### Author Rebuttal · Authors · 2023-08-10

We sincerely thank the PCs, ACs, and all the reviewers for devoting time and effort to this review.

---
We are glad to see that the reviewers are acknowledging that:
- *"The motivation of the method is clear and easy to understand and follow"* (Reviewer JyS2)
- *"The idea of handling spatial and temporal domain divergence separately through disentanglement is interesting"* (Reviewer 3nXx)
- *"Results are comprehensive and well elaborated on multiple datasets, consistent performance improvements over previous methods of single and multi-modal data"* (Reviewer cyvj)
- *"The motivation and writing of this paper are well done"* (Reviewer AobY)
- *"Extensive experiments show the effectiveness of the proposed model"* (Reviewer qCdJ)

---
We have polished the paper, added the experiments and references, and made the clarifications in the revised version. Specifically, we have revised our manuscript to include the following changes according to the reviewers’ insightful comments:
- We have supplemented more ablation experiments, including TranSVAE w/ and w/o disentanglement, to verify the effectiveness of each objective term in our framework.
- We have compared with more SOTA baselines that use multi-modality data sources for video-based UDA, including A3R, CleanAdapt, CycDA, and MixDANN. All the comparison results show that our TransVAE is still a competitive method even using only a single RGB modality.
- We have added complexity analyses on the number of trainable parameters, multiply-accumulate operations, floating-point operations, and inference frame-per-second for our TranSVAE as well as for prior arts.
- For better readability, we have supplemented a notation section for the maths symbols used in the paper.
- We have added missing references suggested by the reviewers.
- We have polished and improved the elaboration of this work.
- We have carefully addressed other comments from the reviewers point by point.

---
We would like to re-emphasize the novelty and technical contributions of this work:
- We provide a new perspective on solving video-based UDA problems, that is from a generative view of cross-domain videos. To the best of our knowledge, this is the first work that tackles the challenging video-based UDA from a domain disentanglement view.
- We develop a generative graphical model for the cross-domain video generation process and propose to utilize the sequential VAE as the base generative model.
- Based on the above generative view, we propose a TranSVAE framework for video-based UDA. By developing four constraints on the latent factors to enable disentanglement to benefit adaptation, the proposed framework is capable of handling the cross-domain divergence from both spatial and temporal levels.
- We conduct extensive experiments on several benchmark datasets to verify the effectiveness of TranSVAE. Comprehensive ablation study also demonstrates the positive effect of each loss term on adaptation.

---
We will actively participate in the Author-Reviewer discussion session. Please don't hesitate to let us know of any additional comments on the manuscript or the changes.

---
Last but not least, we thank the PCs, ACs, and all the reviewers again for the time and effort devoted to this review.

---

### Decision · Program_Chairs · 2023-09-21

**Decision:**

Accept (poster)

**Comment:**

This paper receives four borderline accepts and one weak accept. Initially, the reviewers have common concerns for combinational novelty, insufficient ablation experiment, backbone selection, and recent model result comparison. The rebuttal answers the questions from the reviewers, and provides more results, e.g., a lot of missing comparisons in original submission, recent SOTA comparison, and FLOPs comparison, etc. After the rebuttal, the reviewers' concerns are resolved. AC gives the recommendation of accept (poster) based on the authors' and reviewers' response.